# SIMPLICIAL SMOTE:
# OVERSAMPLING SOLUTION TO THE IMBALANCED LEARNING PROBLEM

## ABSTRACT

SMOTE (Synthetic Minority Oversampling Technique) is the established geometric approach to random oversampling to balance classes in the imbalanced classes learning problem, followed by many extensions. Its idea is to introduce synthetic data points of the minor class, with each new point being the convex combination of an existing data point and one of its $k$-nearest neighbors. This could be viewed as a sampling from the edges of a geometric neighborhood graph. Borrowing tools from the topological data analysis, we propose a generalization of the sampling approach, thus sampling from the simplices of the geometric neighborhood simplicial complex. A new point is defined by the barycentric coordinates concerning a simplex spanned by an arbitrary number of data points being sufficiently close rather than a pair. We evaluate the generalized technique, Simplicial SMOTE, on 23 benchmark datasets and conclude that it outperforms the original SMOTE and its extensions. Moreover, we show how simplicial sampling can be integrated into several popular SMOTE extensions, with our simplicial generalization of Borderline SMOTE further improving the performance on benchmark datasets.

## 1 INTRODUCTION

The imbalanced learning problem is the learning from data when the minority class is dominated by the majority one (He & Garcia, 2009). Many problems in data mining and data analysis are inherently imbalanced in the areas like finance (fraud detection) (Wang et al., 2019; Jiang et al., 2023), marketing (churn prediction) (Liu et al., 2018), medicine (medical diagnosis) (Han et al., 2019), industry (predictive maintenance) (Sridhar & Sanagavarapu, 2021), etc. Often, the rare minority class (a credit fraud, a canceled subscription, the presence of a disease, an equipment failure) is of much more interest than the common majority one. Class imbalance causes the bias of classifiers towards the majority class (Wallace et al., 2011), as the naive classifier assigning all data points to the majority class will achieve an accuracy equal to the majority class proportion.

There are several solutions to overcome this problem: 1) cost-sensitive learning using a class-weighted loss function (Thai-Nghe et al., 2010) or specialized classifier algorithms (Krawczyk, 2016; Esposito et al., 2021), 2) matching the class cardinalities by undersampling the majority class (Hoyos-Osorio et al., 2021), which leads to loss of data, and alternatively 3) oversampling the minority class by introducing new synthetic data points of the minority class (Santoso et al., 2017).

Upsampling techniques include local ones, such as random oversampling (Batista et al., 2004), i.e., randomly duplicating the required number of the minority class instances and global ones, such as Mixup (Zhang et al., 2017) (without labels interpolation), which introduces the new synthetic points as the random convex combinations of randomly selected pairs of the minority class instances. The established oversampling method, Synthetic Minority Oversampling Technique (SMOTE) (Chawla et al., 2002) introduces the new synthetic points as the random convex combinations of pairs consisting of a point and its nearest neighbors. This could be seen as the sampling from the edges of the neighborhood graph.

Neighborhood graphs model the data as the union of $1$-dimensional cells, spanned by pairs of points, which is insufficient to sample from high-dimensional spaces. For example, even for a $2$-dimensional dataset, one could not introduce samples from the entire convex hull spanned by data

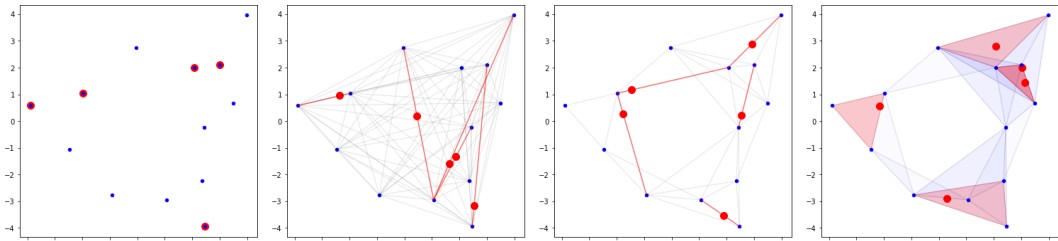

Figure 1: Geometric oversampling algorithms: a) Random oversampling, b) Mixup, c) SMOTE, d) Simplicial SMOTE. Data's graph or simplicial models in blue, selected edges or simplices to sample and sampled synthetic points in red. With no inductive assumptions on data, Random oversampling just duplicates existing points. Assuming that synthetic data points lie within a convex hull of existing points, global methods like Mixup do not respect the intrinsic properties of data such as clusters and holes. While SMOTE, being a local method, improves on this, it still models the data with a union of 1-dimensional cells, unable to sample all of the data's support. Simplicial SMOTE, by modeling data with a union of higher-dimensional simplices, samples dense areas of data's support while avoiding sampling from topological holes.

points using SMOTE. Instead, if we model the data with the union of cells whose dimension is equal to the feature space, for example, a union of 2-dimensional triangles, which are simplices spanned by triples of points, we could sample it.

Therefore, we introduce the generalization of SMOTE, called Simplicial SMOTE, thus sampling from the higher-dimensional simplices of the clique complex of a neighborhood graph. That is, a position of a new point is defined by the barycentric coordinates concerning a simplex spanned by an arbitrary number of data points being sufficiently close, i.e., being in the $p$-ary proximity relation.

OUR CONTRIBUTION

- we propose a novel simplicial extension to the SMOTE algorithm and its modifications, where the new points are sampled from the simplices of the geometric neighborhood simplicial complex,

- we demonstrated that the proposed simplicial extension is orthogonal to various modifications of the SMOTE method by introducing simplicial generalizations of several popular SMOTE variants such as Borderline SMOTE, Safe-level SMOTE, and ADASYN,

- we experimentally demonstrate the the proposed technique is characterized by significant increase in performance for various classifiers and datasets from the `imbalanced-learn` library.

## 2 RELATED WORK

The original SMOTE algorithm introduces synthetic points from the geometric model of the minority class manifold. Several variants of SMOTE instead propose to sample synthetic points from the minority class part of the decision manifold, i.e., the minority points lying on the boundary between classes. The decision manifold is estimated in several ways; Borderline SMOTE (Han et al., 2005) estimates the decision manifold by taking the minority class local density around each minority data point, while SVM SMOTE (Nguyen et al., 2011) first takes the points corresponding to the support vectors of the SVM classifier.

In Safe-level SMOTE (Bunkhumpornpat et al., 2009), a value called safe level ratio is assigned to each edge of the neighborhood graph built over minority class instances, which is the ratio of the numbers of minority class instances for a point $x$ and its neighbor $x'$. If the number of the minority class instances in the neighborhoods of $x$ and $x'$ are zero, no synthetic examples are generated from that edge. Otherwise, a new synthetic sample is a convex combination of the points, and the coefficient depends on the ratio, being close to the minority example with more neighbors of the minority class.

In ADASYN (He et al., 2008), for each minority point, a ratio of majority examples in the neighborhood is computed. The new points are the convex combination of minority class points, with the number of synthetic examples generated using a given minority example being inversely proportional to that ratio.

Several SMOTE extension use clustering schemes for oversampling. MWMOTE (Barua et al., 2012) first identifies the hard-to-learn informative minority class samples and then generates the synthetic samples from the weighted informative minority class samples using a clustering approach. In AHC (Cohen et al., 2006), the minority class is clustered using agglomerative hierarchical clustering, obtaining several clusters at different levels of resulting dendrograms. The new points are the centroids of the clusters, points in the interior of the polytopes spanned by an arbitrary number of points, with the distance less or equal to the chosen histogram level.

In Density-based SMOTE (DBSMOTE) (Bunkhumpornpat et al., 2012), minority class examples are partitioned into disjointed clusters by the DBSCAN algorithm (Ester et al., 1996). The new points are the random convex combinations of two points from the random edge of the shortest path connecting minority points with the pseudo-centroid point, which is the closest to the cluster centroid. LVQ-SMOTE (Nakamura et al., 2013) oversamples the minority class, first approximating is using a set of prototype points obtained by LVQ (Learning Vector Quantization) algorithm De Vries et al. (2016).

Mixup, a geometric method, introduces new synthetic points as a convex combination of a pair of existing points randomly chosen from a dataset (Zhang et al., 2017). Fitting parametric distributions to data, such as the Gaussian distribution, is also used for the minority class oversampling in the imbalanced data classification problem (Xie et al., 2020).

Our work improves the SMOTE modeling and sampling scheme by modeling data with a geometric simplicial complex (Boissonnat et al., 2018; Dey & Wang, 2022), which is the higher-dimensional generalization of a graph. Contrary to global methods such as Mixup or fitting Gaussian distribution, it respects local topological features of data such as clusters and topological holes. The new sampling scheme is orthogonal to most SMOTE improvements and could be used cooperatively with them.

## 3 METHODS

Geometric sampling methods assume that a synthetic point combines (several) existing data point(s). When designing such algorithms, one should decide upon 1) a neighborhood size of each data point, ranging from a point itself to all points of the dataset, and 2) several points a synthetic point is a combination of. Neighborhood relations can describe the former, while the latter corresponds to the relation arity. While popular sampling methods like Mixup or SMOTE model data with a graph, complete or local, based on the binary neighborhood relations, our choice is to model the data with a simplicial complex based on neighborhood relations of arity greater than 2.

Consider a complete graph $H_n$ with a vertex set $X \in \mathbb{R}^d$ of cardinality $n$. A *neighborhood graph* $G = (X, E) \subseteq H_n$ is a subgraph of $H_n$ such that the edge set $E \subseteq \binom{n}{2}$ is instantiated according to a relation $R$ defining a neighborhood of each point

$$\mathcal{N}(x) = \{x' \mid xRx'\}. \tag{1}$$

For example, let $X$ be endowed with a distance function $d : X \times X \to \mathbb{R}_+$. A (symmetrized) $k$-nearest neighbor relation $R^{kNN}$ on $X$ defining a *k-nearest neighbors neighborhood graph* parameterized by $k \in \mathbb{N} \setminus \{0\}$ is

$$R^{kNN}(k) = \left\{ (x, y) \mid d(x, y) \leq \min_k d(x, z), \ z \in X \right\}, \tag{2}$$

where $\min_k(\cdot)$ denotes the $k$-th minimum, hence $\arg\min_k(x, z)$ is the $k$-th neighbor of $x$.

An $\varepsilon$-ball relation $R^\varepsilon$ on $X$ defining the *$\varepsilon$-ball neighborhood graph* given a scale parameter $\varepsilon \in \mathbb{R}_{\geq 0}$ is

$$R^\varepsilon(\varepsilon) = \left\{ (x, y) \mid d(x, y) \leq \varepsilon \right\}, \tag{3}$$

meaning that balls $B_x(\varepsilon/2) \cap B_{x'}(\varepsilon/2) \neq \emptyset$ of radius $\varepsilon/2$ centered at $x$ and $x'$ intersect.

Given a binary relation $R$, a *p-ary relation* $P$ of $R$ is defined as a subset $Y$ of $X$ of cardinality $p$ such that $Y \times Y \subseteq R$, that is, a set $Y$ iff $xRx'$ for any pair $(x, x') \in Y$. A *maximal p-ary relation* $C$ of $R$ defined as a subset $Y$ of $X$ of cardinality $p$ that is maximal concerning inclusion (Henry, 2011).

Whether a binary neighborhood relation corresponds to an edge in a neighborhood graph, a $p$-ary relation corresponds to a graph's $(p + 1)$-clique, more generally, a $p$-simplex in a neighborhood simplicial complex over the vertex set $X$. Points can belong to more than one simplex. All simplices containing a point $x$ are subsets of its neighborhood $\mathcal{N}(x)$ (Henry, 2011).

## 3.1 CLASSIFICATION OF GEOMETRIC SAMPLING METHODS

We classify the existing geometric sampling methods based on the neighborhood and the arity of the neighborhood relation, summarizing them in Table 1.

Table 1: Classification of geometric sampling methods.

|  | Neighborhood size | Relation arity |
|---|---|---|
| Random | 1 | 1 |
| Mixup | $n$ | 2 |
| SMOTE | $k$ | 2 |
| Simplicial SMOTE | $k$ | $[3, k]$ |

For random oversampling, a neighborhood of each point is only the point itself, resulting in synthetic points being duplicates of existing data points.

For Mixup, a neighborhood of each point is global, resulting in the complete graph as the data model, with synthetic points as convex combinations of pairs of data points sampled from the edges of the complete graph.

For SMOTE, a neighborhood of each point is local, defined via a kNN or $\varepsilon$-ball neighborhood relation, resulting in a neighborhood subgraph of the complete graph as the data model. Synthetic points are convex combinations of pairs of data points from the neighborhood graph edges.

$$\hat{\mathbf{x}}_{\mathbf{x}_0, \mathbf{x}_1}(\alpha) = \alpha \mathbf{x}_0 + (1 - \alpha)\mathbf{x}_1. \tag{4}$$

## 3.2 SIMPLICIAL SMOTE

We classified the existing approaches to geometric data modeling and sampling based on the neighborhood relation size and arity. We propose a simple yet effective generalization of SMOTE by considering a general $k$-ary proximity relation. That is, instead of binary relation leading to the proximity graphs, we believe the $k$-ary relation leads to proximity simplicial complex, resulting in the high-dimensional data model, contrary to a graph which is locally 1-dimensional.

Given a set $X$, an *(abstract) simplicial complex* $K$ is a collection of subsets of $X$ called *simplices* such that if a simplex $\sigma$ is in $K$, then all of its subsets $\tau \subseteq \sigma$ are also in $K$. Let $\Lambda_p$ be the set of all vectors of $p + 1$ elements, such that $\lambda_i \geq 0$ and $\sum_{i=0}^{k} \lambda_i = 1$. Given a set of $p + 1$ points $\{\mathbf{x}_i\}_{i=0}^{p}$ in an $d$-dimensional Euclidean space a *geometric p-simplex* $\sigma$ is defined

$$\sigma_{\mathbf{x}_0, \ldots, \mathbf{x}_p} = \left\{ \sum_{i=0}^{p} \lambda_i \mathbf{x}_i \;\middle|\; \boldsymbol{\lambda} \in \Lambda_p \right\}. \tag{5}$$

We call the elements of $\boldsymbol{\lambda}$ *barycentric coordinates* w.r.t. the points spanning a simplex. Barycentric coordinates could be mapped into Euclidean coordinates, resulting in a synthetic point.

$$\hat{\mathbf{x}}_{\mathbf{x}_0, \ldots, \mathbf{x}_p}(\boldsymbol{\lambda}) = \lambda_0 \mathbf{x}_0 + \cdots + \lambda_k \mathbf{x}_p. \tag{6}$$

An example of a geometric complex is the *Vietoris-Rips complex* parameterized by the scale parameter $\varepsilon \in \mathbb{R}_{\geq 0}$

$$K^{VR}(\varepsilon) = \left\{ (x_0, \ldots, x_p) \mid d(x_i, x_j) \leq \varepsilon, \, \forall i, j \right\}. \tag{7}$$

More generally, with any graph, its clique complex could be associated by *expansion*, that is, a complex $K(G)$ has the same vertices and edges $G$, and $(k+1)$-cliques of $G$ are simplices of $K(G)$. The Vietoris-Rips complex is the clique complex of an $\varepsilon$-ball neighborhood graph.

Simplicial complexes can be huge as they grow exponentially w.r.t. $p$, so one can work with its smaller subcomplexes. A *p-skeleton* $L$ of a simplicial complex $K$ is the subcomplex of $K$ with the dimension of simplices at most $p$. Algorithmically, this corresponds to finding cliques up to dimension $p + 1$ instead of maximal cliques.

We outline the Simplicial SMOTE method in Algorithm 1. Note that it has only two hyperparameters, namely the neighborhood size ($k$ for kNN neighborhood graph) and the maximal arity of neighborhood relation (with $p = $ "maximal" by default).

---

**Algorithm 1:** Proposed Simplicial SMOTE

---

**Input**         : Minority class points $X^+$.
**Parameters :** A neighborhood graph $G$ parameters $\theta$,
                  maximal simplex dimension $p$,
**Output**      : Synthetic minority class points $\hat{X}^+$.

1 Construct a neighborhood graph $G_\theta(X^+)$, either
    a) a $k$-NN neighborhood graph, $\theta = k$, or
    b) an $\varepsilon$-ball neighborhood graph, $\theta = \varepsilon$.
2 Compute a full clique complex or its $p$-skeleton $(K_p \circ G_\theta)(X^+)$, eigher finding a) maximal
    cliques $\Sigma_{max}$, $p = $ "maximal", or
    b) cliques up to dimension $(p + 1)$ $\Sigma_p$, $p \in \mathbb{R}_{\geq 2}$.
3 Sample from $\Sigma_{max}$ or $\Sigma_p$ $m = n^- - n^+$ maximal simplices of dimension $p_i$,
    $\Sigma = \{\sigma_i^{(p_i)}\}_{i \in 1, \cdots, m}$.
4 **for** $i \in 1, \ldots, m$ **do**
5        Sample barycentric coordinates $\boldsymbol{\lambda}_i \sim \text{Dir}(\boldsymbol{\alpha})$, where $\boldsymbol{\alpha} = (1, \ldots, 1) \in \mathbb{R}^{(p+1)}$.
6        Compute Euclidean coordinates $\hat{\mathbf{x}}_i = \boldsymbol{\lambda}_i^T \mathbf{X}_i$ w.r.t. a simplex $\sigma_i^{(p_i)} = (\mathbf{x}_0, \ldots, \mathbf{x}_{p_i}) \in \Sigma$ of
         dimension $p_i$.
7 **return** $\{\hat{\mathbf{x}}_i\}_{i \in 1, \ldots, m}$

---

### 3.3 Simplicial generalizations of SMOTE variants

Consider a dataset $\mathcal{X} = \{\mathbf{x}_i, y_i\}_{i \in 1, \ldots, n}$, where $\mathbf{x}_i \in \mathbb{R}^n$ and $y_i \in \{0, 1\}$. By convention, we denote the *minority class* $\mathcal{X}^+ = \{\mathbf{x}_i \mid y_i = 1\}_{i \in 1, \ldots, n^+}$ as positive, and the *majority class* $\mathcal{X}^- = \{\mathbf{x}_j \mid y_i = 0\}_{i \in 1, \ldots, n^-}$ as negative of sizes $n^+ < n^-$ respectively, with $n = n^+ < n^-$.

We denote the *minority neighborhood* of $\mathbf{x}_i$ are the points of the minority class within a given neighborhood of a point $\mathcal{N}^+(\mathbf{x}_i) = \{\mathbf{x}_j \mid \mathbf{x}_i \sim \mathbf{x}_j, y_j = 1\}_{j \in 1, \ldots, k^+}$, and the *majority neighborhood* of $\mathbf{x}_i$ as $\mathcal{N}^-(\mathbf{x}_i) = \{\mathbf{x}_j \mid \mathbf{x}_i \sim \mathbf{x}_j, y_j = 0\}_{j \in 1, \ldots, k^-}$ of sizes $k^+$ and $k^-$ respectively. The majority and minority points ratios within a given neighborhood are defined as $\Delta^+(\mathbf{x}_i) = k^+/k$ and $\Delta^-(\mathbf{x}_i) = k^-/k$ respectively. Given two points from a majority class $\mathbf{x}, \mathbf{x}'$, the *safe-level ratio* is defined

$$\nabla(\mathbf{x}, \mathbf{x}') = \frac{\Delta^+(\mathbf{x})}{\Delta^+(\mathbf{x}')} = \frac{k^+(\mathbf{x})}{k(\mathbf{x})} \frac{k(\mathbf{x}')}{k^+(\mathbf{x}')}. \tag{8}$$

The original SMOTE algorithm constructs the minority neighborhood graph and samples points from its edges without considering the majority class. Several variants of the SMOTE algorithm improve reinforcing the points close to the boundary between types by considering the density of the majority class relative to the points from the minority.

- Borderline SMOTE hard modifies the minority neighborhood graph, setting zero probability to sample synthetic points based on only points outside the borderline subset (i.e., the union of safe and noise points). The probability of sample points from simplices spanned by borderline and borderline and outside points remains the same.

- ADASYN hard/soft modifies the minority neighborhood graph, zeroing/lowering probabilities to sample synthetic points using points in the safe region. The probability of sample points from simplices changes proportionally to their safety/homogeneity ratio (the higher the homogeneity - the lower the probability).

- Safe-level SMOTE leaves the minority neighborhood graph intact but modifies the barycentric coordinate weights on simplices, adding more weight to sample synthetic points using points with a higher safety/homogeneity ratio.

We generalize the SMOTE algorithm variants to use the simplicial sampling scheme.

### 3.3.1 SIMPLICIAL BORDERLINE SMOTE

The extension assumption is that the examples on the borderline and the ones nearby are more apt to be misclassified than the ones far from the borderline and, thus, more important for classification. The examples far from the borderline may contribute little to classification results.

The *borderline subset* of the minority class $B(\mathcal{X}^+)$ is defined

$$B(\mathcal{X}^+) = \left\{ \mathbf{x}_i, y_i = 0 \mid \frac{|\mathcal{N}^+(\mathbf{x}_i)|}{|\mathcal{N}(\mathbf{x}_i)|} < 1/2, |\mathcal{N}^+(\mathbf{x}_i)| \neq 0 \right\} \tag{9}$$

that is the points whose the larger part of the nearest neighbors belong to the majority class, except those whose nearest neighbors are completely majority class instances and are considered noise. The new points are the convex combination of the simplices of a simplicial complex built upon the borderline points and their nearest neighbors from the minority class.

### 3.3.2 SIMPLICIAL SAFE-LEVEL SMOTE

The original SMOTE algorithm considers sampling from a $k$-simplex according to the Dirichlet distribution $\mathrm{Dir}(\boldsymbol{\alpha})$, where $\boldsymbol{\alpha} \in \mathbb{R}^k_{>0}$. Without any further assumptions, the distribution is symmetric, i.e., all of the vector $\boldsymbol{\alpha}$ elements have the same value (usually $1$, resulting in the uniform distribution on a simplex). Safe-level SMOTE (Bunkhumpornpat et al., 2009) modifies the elements of $\boldsymbol{\alpha}$ by setting them based on the ratio of majority neighborhood ratios, resulting in synthetic points to be generated closer to safer minority points, i.e., having a larger proportion of neighbors of the same class. A simplicial generalization is to set the parameter $\alpha_i = 1 + \Delta^+(x_i)$.

### 3.3.3 SIMPLICIAL ADASYN

While Borderline SMOTE answers the question from which simplex to sample, selecting simplices spanned by borderline points, and Safe-level SMOTE answers the question from where precisely on a simplex to sample sampling closer to safer points, ADASYN answers the question of how much to sample from a simplex, inversely proportional to the average safety of points. Therefore, its simplicial generalization is to average arbitrary safety values instead of just a pair.

### 3.4 COMPLEXITY ANALYSIS

Algorithm complexity depends on complexity of the neighborhood graph construction and expansion. The naive nearest neighbor search have complexity of $O(n^2)$, while approximated nearest neighbor search lowers it to $O(n)$.

Enumerating all maximal cliques in a graph with $n$ vertices and $m$ edges is an NP-complete problem, requiring exponential time in the worst case. Up to $n^{n/3}$ maximal cliques exist in a graph with $n$ vertices (Moon & Moser, 1965). Yet, as the neighborhood graphs are sparse, the various bounds were given regarding the number of edges, node degree, and arboricity of a graph. In a graph with maximum degree $\delta$ the time complexity of MCE is $O(\delta^4)$ per clique and $O((n - \delta)3^{\delta/3}\delta^4)$ total

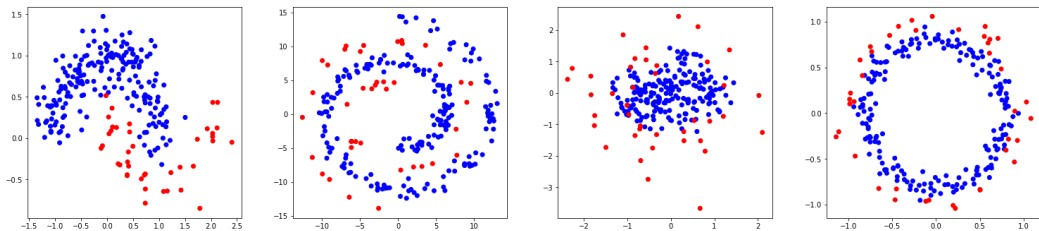

Figure 2: Synthetic data: a) moons, b) swiss rolls, c) a Gaussian inside a sphere, d) a sphere inside a sphere.

(Makino & Uno, 2004; Eppstein et al., 2013). The *arboricity* is the minimum number of edge-disjoint spanning forests into which the graph can be decomposed. For a graph of arboricity $a$, the complexity of MCE is $O(am)$ (Chiba & Nishizeki, 1985).

Graph expansion could also be done via maximal clique enumeration (MCE). It is shown that the time complexity of MCE is $O(m)$, thus linear in the number of edges (Yu & Liu, 2017; 2019). Enumeration of all cliques up to the size $p$ can be done in either inductive, incremental, or top-down enumeration approach after solving the MCE problem (Zomorodian, 2010). Recently, an algorithm conjectured to be optimal was proposed for this task, which is on average was shown to be approximately a magnitude faster than the incremental algorithm in practice Rieser (2023).

## 4 RESULTS

### 4.1 SYNTHETIC DATA

First, we evaluated the proposed method and its alternatives, in comparison with the original SMOTE (Chawla et al., 2002), sampling from a fitted Gaussian (Xie et al., 2020), random oversampling (Batista et al., 2004), and the Mixup (Zhang et al., 2017) algorithms, on several model datasets to emphasize the importance of modeling the data locally, as well as the advantage of the simplicial complex data model.

As the model data, we have generated the following datasets partially using the `scikit-learn` library (Pedregosa et al., 2011), shown in Figure 2: moons, swiss rolls, a Gaussian inside a circle, a circle inside a circle. As model datasets have complex geometric structures, global methods are conjectured to underperform by generating synthetic points of minority class within the support of the majority class.

Table 2: Synthetic data classification results.

|  | Imbalanced | Gaussian | Random | Mixup | SMOTE | Simplicial |
|---|---|---|---|---|---|---|
| moons | 0.9511 | 0.8830 | 0.9485 | 0.9348 | **0.9694** | **0.9694** |
| swiss_rolls | 0.5317 | 0.6673 | 0.7168 | 0.6774 | **0.7208** | 0.6823 |
| g_circle | 0.7129 | 0.6750 | 0.7089 | 0.6542 | 0.6937 | **0.7269** |
| circles | 0.6541 | 0.7060 | 0.6777 | 0.6356 | 0.7005 | **0.7139** |
| **rank** | 4.0000 | 4.5000 | 3.2500 | 5.2500 | 2.3750 | **1.6250** |

All model datasets consist of $n = 350$ points, with the size of the minority class $n^+ = 50$ (shown in red) and the size of the majority class $n^- = 300$ (shown in blue), with class imbalance ratio of 6.

For SMOTE and Simplicial SMOTE, we performed a grid search for the neighborhood size parameter $k$ of the kNN neighborhood graph ranging from 3 to 8 with a step 1. We report the F1 score averaged over 5 runs using 4-fold cross validation in Table 2 for the $k$-nearest neighbors classifier with default hyperparameters from the `scikit-learn` library (Pedregosa et al., 2011).

Results show when data is of complex geometric structure, global methods such as fitting a Gaussian or sampling from a complete graph using Mixup underperform the local techniques such as

SMOTE and Simplicial SMOTE, as well as the simple random oversampling. Simplicial SMOTE has performed the best, achieving the highest rank among all sampling methods, and is generally better than its graph- based counterpart.

Table 3: Classification results on benchmark datasets for the $k$-nearest neighbor classifier. F1 score averaged over 10 runs using 5-fold cross-validation is reported. **Best** and **second-best** ranked results are highlighted.

| | Imbalanced | Random | Mixup | SMOTE | B-SMOTE | Safelevel | ADASYN | MWMOTE | DBSMOTE | LVQ | AHC | Simplicial | B-Simplicial |
|---|---|---|---|---|---|---|---|---|---|---|---|---|---|
| ecoli | 0.5567 | 0.5492 | 0.5965 | 0.5922 | 0.6109 | 0.5871 | 0.5878 | 0.6011 | 0.6192 | 0.5988 | 0.6022 | 0.6382 | 0.6238 |
| optical_digits | 0.9636 | 0.9489 | 0.9414 | 0.9418 | 0.9541 | 0.9432 | 0.9419 | 0.9369 | 0.9491 | 0.9588 | 0.9647 | 0.9482 | 0.9541 |
| satimage | 0.6899 | 0.6535 | 0.5982 | 0.6065 | 0.5964 | 0.5978 | 0.5880 | 0.5749 | 0.6815 | 0.6659 | 0.6527 | 0.5990 | 0.5944 |
| pen_digits | 0.9928 | 0.9901 | 0.9892 | 0.9909 | 0.9928 | 0.9903 | 0.9920 | 0.9896 | 0.9920 | 0.9929 | 0.9919 | 0.9925 | 0.9927 |
| abalone | 0.1853 | 0.3305 | 0.3851 | 0.3520 | 0.3678 | 0.3769 | 0.3463 | 0.3749 | 0.3242 | 0.3117 | 0.3076 | 0.3710 | 0.3728 |
| sick_euthyroid | 0.5575 | 0.5706 | 0.5926 | 0.5754 | 0.5681 | 0.5311 | 0.5626 | 0.5621 | 0.6218 | 0.5688 | 0.6064 | 0.6141 | 0.6087 |
| spectrometer | 0.7730 | 0.8494 | 0.8369 | 0.8568 | 0.8495 | 0.8243 | 0.8426 | 0.8493 | 0.7882 | 0.8345 | 0.8287 | 0.8621 | 0.8490 |
| car_eval_34 | 0.6148 | 0.5505 | 0.5655 | 0.5796 | 0.5800 | 0.6012 | 0.5742 | 0.6517 | 0.5505 | 0.7223 | 0.6358 | 0.6248 | 0.6082 |
| us_crime | 0.3703 | 0.4419 | 0.4454 | 0.4320 | 0.4635 | 0.4484 | 0.4230 | 0.4167 | 0.4419 | 0.4665 | 0.4557 | 0.4536 | 0.4890 |
| yeast_ml8 | 0.0445 | 0.1469 | 0.1601 | 0.1651 | 0.1675 | 0.1625 | 0.1655 | 0.1607 | 0.1469 | 0.1596 | 0.1621 | 0.1631 | 0.1673 |
| scene | 0.0954 | 0.2459 | 0.2513 | 0.2393 | 0.2520 | 0.2524 | 0.2362 | 0.2338 | 0.1110 | 0.2777 | 0.2594 | 0.2271 | 0.2589 |
| libras_move | 0.7182 | 0.8121 | 0.7495 | 0.7815 | 0.7780 | 0.7350 | 0.7754 | 0.7741 | 0.8121 | 0.8066 | 0.7663 | 0.8028 | 0.7847 |
| thyroid_sick | 0.5067 | 0.5246 | 0.5261 | 0.5235 | 0.5309 | 0.4653 | 0.5251 | 0.5283 | 0.5303 | 0.5081 | 0.5520 | 0.5672 | 0.5647 |
| coil_2000 | 0.0454 | 0.1738 | 0.1702 | 0.1744 | 0.1770 | 0.1716 | 0.1738 | 0.1718 | 0.0502 | 0.1184 | 0.1657 | 0.1744 | 0.1770 |
| arrhythmia | 0.0305 | 0.2225 | 0.1859 | 0.1935 | 0.1984 | 0.1647 | 0.2008 | 0.2091 | 0.2225 | 0.1753 | 0.2331 | 0.1931 | 0.1860 |
| solar_flare_m0 | 0.0496 | 0.2243 | 0.2069 | 0.2138 | 0.2398 | 0.2388 | 0.2213 | 0.2063 | 0.0488 | 0.2384 | 0.1854 | 0.2225 | 0.2388 |
| oil | 0.3302 | 0.4614 | 0.4623 | 0.4583 | 0.4833 | 0.4080 | 0.4403 | 0.4336 | 0.4614 | 0.5017 | 0.5140 | 0.5331 | 0.5444 |
| car_eval_4 | 0.0966 | 0.3884 | 0.4886 | 0.4513 | 0.4503 | 0.4141 | 0.4528 | 0.5130 | 0.3884 | 0.7011 | 0.3312 | 0.5658 | 0.5600 |
| wine_quality | 0.1218 | 0.3044 | 0.2143 | 0.2548 | 0.2766 | 0.2521 | 0.2542 | 0.2213 | 0.1459 | 0.2353 | 0.3241 | 0.2580 | 0.2797 |
| letter_img | 0.9712 | 0.9522 | 0.9098 | 0.9403 | 0.9607 | 0.9298 | 0.9543 | 0.9120 | 0.9651 | 0.9677 | 0.9701 | 0.9652 | 0.9608 |
| yeast_me2 | 0.2597 | 0.3260 | 0.2692 | 0.2943 | 0.3637 | 0.3067 | 0.2931 | 0.3111 | 0.3184 | 0.2704 | 0.3764 | 0.3255 | 0.3714 |
| ozone_level | 0.1633 | 0.2328 | 0.2047 | 0.2077 | 0.2308 | 0.2326 | 0.2075 | 0.2089 | 0.2328 | 0.2607 | 0.2861 | 0.2266 | 0.2532 |
| abalone_19 | 0.0000 | 0.0346 | 0.0482 | 0.0448 | 0.0555 | 0.0367 | 0.0450 | 0.0252 | 0.0410 | 0.0337 | 0.0050 | 0.0568 | 0.0598 |
| **rank** | 10.4783 | 7.1739 | 8.5652 | 7.5217 | 4.9130 | 8.5652 | 8.1304 | 8.4783 | 7.4348 | 5.9130 | 5.5652 | **4.5217** | **3.7391** |

Table 4: Classification results on benchmark datasets for the gradient boosting classifier. F1 score averaged over 10 runs using 5-fold cross-validation is reported. **Best** and **second-best** ranked results are highlighted.

| | Imbalanced | Random | Mixup | SMOTE | B-SMOTE | Safelevel | ADASYN | MWMOTE | DBSMOTE | LVQ | AHC | Simplicial | B-Simplicial |
|---|---|---|---|---|---|---|---|---|---|---|---|---|---|
| ecoli | 0.6351 | 0.6125 | 0.6193 | 0.6444 | 0.6226 | 0.6145 | 0.6346 | 0.6055 | 0.5973 | 0.6561 | 0.6450 | 0.6614 | 0.6340 |
| optical_digits | 0.8713 | 0.8550 | 0.9086 | 0.9048 | 0.8888 | 0.8987 | 0.8991 | 0.9031 | 0.8520 | 0.8681 | 0.8879 | 0.9039 | 0.8908 |
| satimage | 0.5857 | 0.5494 | 0.5879 | 0.5799 | 0.5522 | 0.5772 | 0.5436 | 0.5752 | 0.5750 | 0.6144 | 0.6101 | 0.5974 | 0.5821 |
| pen_digits | 0.9471 | 0.9487 | 0.9054 | 0.9562 | 0.9196 | 0.9559 | 0.9235 | 0.9397 | 0.9523 | 0.9425 | 0.9551 | 0.9597 | 0.9219 |
| abalone | 0.0339 | 0.3890 | 0.4250 | 0.3990 | 0.4033 | 0.3947 | 0.3928 | 0.4024 | 0.3976 | 0.3876 | 0.3126 | 0.4139 | 0.4154 |
| sick_euthyroid | 0.8642 | 0.8388 | 0.8415 | 0.8433 | 0.8408 | 0.7852 | 0.8412 | 0.8397 | 0.8726 | 0.8572 | 0.8504 | 0.8557 | 0.8466 |
| spectrometer | 0.7683 | 0.7973 | 0.7778 | 0.8109 | 0.7998 | 0.8234 | 0.8126 | 0.7816 | 0.7897 | 0.7805 | 0.7926 | 0.8350 | 0.8110 |
| car_eval_34 | 0.8870 | 0.8359 | 0.9511 | 0.9254 | 0.8999 | 0.8461 | 0.9172 | 0.9196 | 0.8364 | 0.9013 | 0.9107 | 0.9389 | 0.9385 |
| us_crime | 0.4790 | 0.4946 | 0.5095 | 0.4952 | 0.5138 | 0.5031 | 0.4894 | 0.4919 | 0.4938 | 0.5187 | 0.5158 | 0.5175 | 0.5198 |
| yeast_ml8 | 0.0145 | 0.1226 | 0.1385 | 0.1324 | 0.1403 | 0.1555 | 0.1302 | 0.1267 | 0.1196 | 0.1247 | 0.0011 | 0.1337 | 0.1373 |
| scene | 0.0836 | 0.2938 | 0.2724 | 0.2676 | 0.2808 | 0.2816 | 0.2645 | 0.2575 | 0.1404 | 0.0847 | 0.1830 | 0.2625 | 0.2757 |
| libras_move | 0.6488 | 0.8015 | 0.7913 | 0.7982 | 0.8146 | 0.7828 | 0.7957 | 0.7944 | 0.8222 | 0.8078 | 0.7407 | 0.8058 | 0.7900 |
| thyroid_sick | 0.8431 | 0.8258 | 0.8079 | 0.8316 | 0.8309 | 0.6430 | 0.8274 | 0.7905 | 0.8282 | 0.8237 | 0.8399 | 0.8384 | 0.8420 |
| coil_2000 | 0.0099 | 0.2217 | 0.0060 | 0.1682 | 0.2060 | 0.2321 | 0.0197 | 0.0716 | 0.0097 | 0.0080 | 0.1743 | 0.1062 | 0.1467 |
| arrhythmia | 0.7730 | 0.7799 | 0.7950 | 0.8057 | 0.8043 | 0.5991 | 0.8069 | 0.8053 | 0.7757 | 0.7745 | 0.7900 | 0.8050 | 0.8068 |
| solar_flare_m0 | 0.0996 | 0.1930 | 0.1202 | 0.2080 | 0.2028 | 0.2092 | 0.1989 | 0.1892 | 0.0793 | 0.1417 | 0.1076 | 0.1554 | 0.1964 |
| oil | 0.4772 | 0.5593 | 0.5806 | 0.5727 | 0.5675 | 0.4988 | 0.5671 | 0.5494 | 0.5529 | 0.5618 | 0.5287 | 0.5884 | 0.5845 |
| car_eval_4 | 0.9529 | 0.8257 | 0.9487 | 0.8998 | 0.8918 | 0.8418 | 0.8950 | 0.8769 | 0.8277 | 0.9123 | 0.9470 | 0.9178 | 0.9185 |
| wine_quality | 0.1939 | 0.2746 | 0.2164 | 0.2475 | 0.2796 | 0.2554 | 0.2443 | 0.2342 | 0.2175 | 0.2912 | 0.2467 | 0.2439 | 0.2717 |
| letter_img | 0.8621 | 0.6956 | 0.8541 | 0.7882 | 0.7543 | 0.7559 | 0.7571 | 0.7973 | 0.8042 | 0.8448 | 0.8663 | 0.8523 | 0.8063 |
| yeast_me2 | 0.2956 | 0.3314 | 0.3124 | 0.3311 | 0.3717 | 0.3108 | 0.3227 | 0.3496 | 0.3525 | 0.3325 | 0.3649 | 0.3556 | 0.3988 |
| ozone_level | 0.1458 | 0.3422 | 0.3031 | 0.3151 | 0.3552 | 0.2984 | 0.3227 | 0.3174 | 0.3442 | 0.3184 | 0.2476 | 0.3404 | 0.3543 |
| abalone_19 | 0.0000 | 0.0479 | 0.0503 | 0.0550 | 0.0774 | 0.0529 | 0.0570 | 0.0542 | 0.0544 | 0.0530 | 0.0000 | 0.0740 | 0.0913 |
| **rank** | 9.5000 | 8.4348 | 7.1304 | 5.5217 | 5.6957 | 7.8696 | 7.1739 | 8.2609 | 8.4348 | 7.0870 | 7.5000 | **4.0435** | **4.3478** |

## 4.2 REAL DATA

In the following, we evaluated the proposed Simplicial SMOTE and its simplicial Borderline extension, comparing it to random oversampling, Mixup (without label smoothing), and several SMOTE variants from the `imbalanced-learn` (Lemaître et al., 2017) and `smote-variants` (Kovács, 2019) libraries. Evaluation was done on 23 benchmark datasets from the `imbalanced-learn` library. All datasets contain two data classes and are summarized in the appendix A. The class imbalance ratio ranges from 9 to 130. Data dimensionality ranges from 6 to 617. Each dataset was normalized to zero mean and unit variance.

We performed a grid search for the neighborhood size parameter $k$ of the kNN neighborhood graph and the maximal simplex dimension $p$ for SMOTE-based methods and (Borderline) Simplicial SMOTE, respectively. The neighborhood size $k$ ranged from 3 to $\text{ceil}(\sqrt[3]{n^+} + \log d)$ with a step 2, where $n^+$ is the minority class size and $d$ is the dimension of the dataset. The maximal simplex dimension $p$ ranged from 3 to $k$, with a step 1.

We report the F1 score and the Matthew's correlation coefficient in averaged over 10 runs using 5-fold cross-validation for 3 classifiers – $k$-nearest neighbors, gradient boosting, and multilayer perceptron from the `scikit-learn` library (Pedregosa et al., 2011). We provide F1 score tables for $k$-nearest neighbors, gradient boosting classifiers in the results section, for the additional experiments, refer to the appendix B. We used default hyperparameters for $k$-nearest neighbor classifier, set maximum tree depth to 2 for gradient boosting. For multilayer perception, we reduced the hidden layer's size to 32 and increased the maximum number of iterations to 500.

Classification results on benchmark imbalanced datasets show the advantage of the proposed method over its counterparts, including the original SMOTE in terms of F1 and Matthew's correlation coefficient scores. The former one is the metric of choice when one is interested most in the correct classification in the minority class over the majority one (Davis & Goadrich, 2006). The latter considers all cells of the contingency table Chicco & Jurman (2020; 2023).

We performed statistical significance testing of the performance Simplicial SMOTE and Borderline SMOTE over original SMOTE using Wilcoxon signed-rank test. For all classifiers and metrics the number of significant wins over datasets is greater than number os losses. Refer to the appendix C for p-values and the information on wins, losses, and draws.

## 5 CONCLUSIONS

In our work, we classified the existing approaches to geometric data modeling and sampling based on the neighborhood relation size and arity. We proposed a new instance of geometric oversampling called Simplicial SMOTE. As the original SMOTE algorithm, it models data locally by the neighborhood size $k$ much less than the amount $n$ of data points. Yet, instead of a graph model of data, which sample synthetic points as random convex combinations from the neighborhood graph edges, it uses a simplicial complex to model the data to sample synthetic points as random convex combinations from its simplices, formed by points being in $p$-ary neighborhood relation.

Experimentally, we have shown on model and real imbalanced datasets that this approach to data modeling and sampling performs better than several sampling methods, including Mixup, original SMOTE and several its popular variants, to solve the classification problem in the presence of data imbalance.

In our experiments, we have concluded that the best number of points to span a simplex generally is not maximal, as the synthetic points, which are a convex combination of a large number of existing data points, could be potentially either too similar for small neighborhood size, or over smoothed for the large one. Thus, we recommend doing a grid search over the maximal simplex dimension $p$ instead of specifying it as maximal by inclusion.

As our method improves the original SMOTE algorithm only in terms of sampling, we have shown that it is orthogonal and compatible with one of the most popular SMOTE variants, namely Borderline SMOTE, Safe-level SMOTE, and ADASYN. We have experimentally evaluated Simplicial Borderline SMOTE, showing its increased performance relative to both original SMOTE and Borderline SMOTE.

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

## A  DATASETS PROPERTIES

Table 5: Benchmark datasets and their properties.

|  | Features | Size | Minor | Major | Ratio |
|---|---|---|---|---|---|
| ecoli | 7 | 336 | 35 | 301 | 9 |
| optical_digits | 64 | 5620 | 554 | 5066 | 10 |
| satimage | 36 | 6435 | 626 | 5809 | 10 |
| pen_digits | 16 | 10992 | 1055 | 9937 | 10 |
| abalone | 10 | 4177 | 391 | 3786 | 10 |
| sick_euthyroid | 42 | 3163 | 293 | 2870 | 10 |
| spectrometer | 93 | 531 | 45 | 486 | 11 |
| car_eval_34 | 21 | 1728 | 134 | 1594 | 12 |
| us_crime | 100 | 1994 | 150 | 1844 | 13 |
| yeast_ml8 | 103 | 2417 | 178 | 2239 | 13 |
| scene | 294 | 2407 | 177 | 2230 | 13 |
| libras_move | 90 | 360 | 24 | 336 | 14 |
| thyroid_sick | 52 | 3772 | 231 | 3541 | 16 |
| coil_2000 | 85 | 9822 | 586 | 9236 | 16 |
| arrhythmia | 278 | 452 | 25 | 427 | 18 |
| solar_flare_m0 | 32 | 1389 | 68 | 1321 | 20 |
| oil | 49 | 937 | 41 | 896 | 22 |
| car_eval_4 | 21 | 1728 | 65 | 1663 | 26 |
| wine_quality | 11 | 4898 | 183 | 4715 | 26 |
| letter_img | 16 | 20000 | 734 | 19266 | 27 |
| yeast_me2 | 8 | 1484 | 51 | 1433 | 29 |
| ozone_level | 72 | 2536 | 73 | 2463 | 34 |
| abalone_19 | 10 | 4177 | 32 | 4145 | 130 |

## B ADDITIONAL EXPERIMENTS

Table 6: Classification results on benchmark datasets for the $k$-nearest neighbor classifier. Matthew's correlation coefficient averaged over 10 runs using 5-fold cross-validation is reported. **__Best__** and **second-best** ranked results are highlighted.

| | Imbalanced | Random | Mixup | SMOTE | B-SMOTE | Safelevel | ADASYN | MWMOTE | DBSMOTE | LVQ | AHC | Simplicial | B-Simplicial |
|---|---|---|---|---|---|---|---|---|---|---|---|---|---|
| ecoli | 0.5328 | 0.5132 | 0.5721 | 0.5659 | 0.5840 | 0.5622 | 0.5630 | 0.5713 | 0.5865 | 0.5739 | 0.5660 | 0.6110 | 0.5955 |
| optical_digits | 0.9601 | 0.9437 | 0.9359 | 0.9363 | 0.9494 | 0.9378 | 0.9362 | 0.9312 | 0.9439 | 0.9544 | 0.9610 | 0.9431 | 0.9494 |
| satimage | 0.6585 | 0.6340 | 0.5890 | 0.5978 | 0.5861 | 0.5817 | 0.5704 | 0.5820 | 0.6499 | 0.6304 | 0.6302 | 0.5909 | 0.5851 |
| pen_digits | 0.9921 | 0.9891 | 0.9881 | 0.9899 | 0.9920 | 0.9893 | 0.9911 | 0.9885 | 0.9912 | 0.9922 | 0.9911 | 0.9917 | 0.9920 |
| abalone | 0.1450 | 0.2585 | 0.3278 | 0.3023 | 0.3149 | 0.3384 | 0.2955 | 0.3290 | 0.2491 | 0.2344 | 0.2270 | 0.3157 | 0.3168 |
| sick_euthyroid | 0.5430 | 0.5400 | 0.5592 | 0.5483 | 0.5378 | 0.4983 | 0.5369 | 0.5345 | 0.5827 | 0.5323 | 0.5662 | 0.5754 | 0.5738 |
| spectrometer | 0.7784 | 0.8410 | 0.8267 | 0.8472 | 0.8408 | 0.8129 | 0.8324 | 0.8405 | 0.7904 | 0.8265 | 0.8259 | 0.8537 | 0.8388 |
| car_eval_34 | 0.6274 | 0.5492 | 0.5855 | 0.5927 | 0.5876 | 0.6054 | 0.5883 | 0.6599 | 0.5492 | 0.7160 | 0.6085 | 0.6294 | 0.6160 |
| us_crime | 0.3809 | 0.4006 | 0.4309 | 0.4215 | 0.4496 | 0.4365 | 0.4146 | 0.4047 | 0.4006 | 0.4257 | 0.4146 | 0.4370 | 0.4642 |
| yeast_ml8 | 0.0763 | 0.0533 | 0.0851 | 0.0879 | 0.0862 | 0.0820 | 0.0908 | 0.0865 | 0.0533 | 0.0665 | 0.0861 | 0.0988 | 0.0949 |
| scene | 0.1324 | 0.1777 | 0.2324 | 0.2162 | 0.2222 | 0.2008 | 0.2146 | 0.2094 | 0.1338 | 0.2161 | 0.2362 | 0.2010 | 0.2256 |
| libras_move | 0.7374 | 0.8067 | 0.7496 | 0.7805 | 0.7767 | 0.7314 | 0.7756 | 0.7698 | 0.8067 | 0.8001 | 0.7632 | 0.7989 | 0.7795 |
| thyroid_sick | 0.5278 | 0.5014 | 0.5044 | 0.5061 | 0.5100 | 0.4511 | 0.5086 | 0.5127 | 0.5103 | 0.5040 | 0.5279 | 0.5405 | 0.5406 |
| coil_2000 | 0.0497 | 0.1097 | 0.1076 | 0.1147 | 0.1150 | 0.1160 | 0.1141 | 0.1108 | 0.0554 | 0.0815 | 0.1039 | 0.1134 | 0.1152 |
| arrhythmia | 0.0271 | 0.1801 | 0.1685 | 0.1794 | 0.1830 | 0.1152 | 0.1904 | 0.1874 | 0.1801 | 0.1231 | 0.2089 | 0.1931 | 0.1739 |
| solar_flare_m0 | 0.0486 | 0.1820 | 0.1695 | 0.1754 | 0.2003 | 0.2007 | 0.1823 | 0.1667 | 0.0443 | 0.1975 | 0.1393 | 0.1845 | 0.1992 |
| oil | 0.3956 | 0.4462 | 0.4573 | 0.4552 | 0.4758 | 0.3929 | 0.4391 | 0.4227 | 0.4462 | 0.4851 | 0.5028 | 0.5209 | 0.5316 |
| car_eval_4 | 0.1646 | 0.4052 | 0.5447 | 0.5101 | 0.5093 | 0.4746 | 0.5129 | 0.5525 | 0.4052 | 0.7192 | 0.3259 | 0.6016 | 0.5965 |
| wine_quality | 0.1779 | 0.2852 | 0.2304 | 0.2543 | 0.2667 | 0.2408 | 0.2529 | 0.2324 | 0.1366 | 0.2126 | 0.3003 | 0.2573 | 0.2720 |
| letter_img | 0.9702 | 0.9512 | 0.9100 | 0.9396 | 0.9597 | 0.9292 | 0.9534 | 0.9121 | 0.9638 | 0.9665 | 0.9692 | 0.9641 | 0.9598 |
| yeast_me2 | 0.3104 | 0.3193 | 0.2987 | 0.3109 | 0.3603 | 0.3108 | 0.3093 | 0.3166 | 0.3283 | 0.2768 | 0.3598 | 0.3364 | 0.3682 |
| ozone_level | 0.2031 | 0.2223 | 0.2483 | 0.2462 | 0.2637 | 0.2475 | 0.2435 | 0.2400 | 0.2223 | 0.2614 | 0.2788 | 0.2593 | 0.2715 |
| abalone_19 | -0.0001 | 0.0269 | 0.0704 | 0.0660 | 0.0586 | 0.0318 | 0.0622 | 0.0176 | 0.0338 | 0.0425 | 0.0026 | 0.0807 | 0.0630 |
| **rank** | 9.4783 | 8.4348 | 8.0000 | 6.7826 | 5.2174 | 8.8261 | 7.7391 | 8.1304 | 8.2174 | 6.6087 | 6.0435 | **3.7826** | **__3.7391__** |

Table 7: Classification results on benchmark datasets for the gradient boosting classifier. Matthew's correlation coefficient averaged over 10 runs using 5-fold cross-validation is reported. **__Best__** and **second-best** ranked results are highlighted.

| | Imbalanced | Random | Mixup | SMOTE | B-SMOTE | Safelevel | ADASYN | MWMOTE | DBSMOTE | LVQ | AHC | Simplicial | B-Simplicial |
|---|---|---|---|---|---|---|---|---|---|---|---|---|---|
| ecoli | 0.6220 | 0.5746 | 0.5838 | 0.6134 | 0.5847 | 0.5817 | 0.6016 | 0.5681 | 0.5633 | 0.6281 | 0.6192 | 0.6307 | 0.5975 |
| optical_digits | 0.8650 | 0.8427 | 0.8994 | 0.8948 | 0.8769 | 0.8879 | 0.8884 | 0.8930 | 0.8395 | 0.8581 | 0.8803 | 0.8939 | 0.8793 |
| satimage | 0.5600 | 0.5293 | 0.5522 | 0.5510 | 0.5288 | 0.5501 | 0.5232 | 0.5462 | 0.5448 | 0.5750 | 0.5686 | 0.5576 | 0.5504 |
| pen_digits | 0.9430 | 0.9437 | 0.8954 | 0.9516 | 0.9116 | 0.9514 | 0.9156 | 0.9340 | 0.9473 | 0.9372 | 0.9512 | 0.9556 | 0.9141 |
| abalone | 0.0523 | 0.3669 | 0.3678 | 0.3733 | 0.3732 | 0.3732 | 0.3715 | 0.3658 | 0.3699 | 0.3543 | 0.2580 | 0.3710 | 0.3697 |
| sick_euthyroid | 0.8510 | 0.8253 | 0.8267 | 0.8296 | 0.8268 | 0.7703 | 0.8275 | 0.8252 | 0.8602 | 0.8431 | 0.8355 | 0.8414 | 0.8320 |
| spectrometer | 0.7665 | 0.7853 | 0.7642 | 0.7976 | 0.7877 | 0.8103 | 0.8004 | 0.7669 | 0.7822 | 0.7652 | 0.7851 | 0.8279 | 0.7997 |
| car_eval_34 | 0.8819 | 0.8338 | 0.9478 | 0.9220 | 0.8966 | 0.8436 | 0.9139 | 0.9157 | 0.8343 | 0.8960 | 0.9053 | 0.9352 | 0.9346 |
| us_crime | 0.4637 | 0.4697 | 0.4724 | 0.4621 | 0.4815 | 0.4824 | 0.4584 | 0.4581 | 0.4690 | 0.4859 | 0.4862 | 0.4825 | 0.4840 |
| yeast_ml8 | 0.0167 | 0.0259 | 0.0413 | 0.0304 | 0.0471 | 0.0587 | 0.0277 | 0.0264 | 0.0224 | 0.0434 | -0.0063 | 0.0431 | 0.0495 |
| scene | 0.1244 | 0.2346 | 0.2095 | 0.2045 | 0.2177 | 0.2256 | 0.2007 | 0.1910 | 0.1772 | 0.1106 | 0.2120 | 0.1970 | 0.2122 |
| libras_move | 0.6801 | 0.8103 | 0.7928 | 0.7994 | 0.8149 | 0.7788 | 0.7972 | 0.8014 | 0.8286 | 0.8070 | 0.7587 | 0.8122 | 0.7901 |
| thyroid_sick | 0.8350 | 0.8205 | 0.7987 | 0.8265 | 0.8255 | 0.6493 | 0.8226 | 0.7820 | 0.8181 | 0.8139 | 0.8305 | 0.8287 | 0.8327 |
| coil_2000 | 0.0209 | 0.1992 | 0.0156 | 0.1391 | 0.1559 | 0.1847 | 0.1417 | 0.0370 | 0.0735 | 0.0305 | 0.0294 | 0.1093 | 0.1271 |
| arrhythmia | 0.7696 | 0.7780 | 0.7921 | 0.8041 | 0.8014 | 0.5841 | 0.8051 | 0.8028 | 0.7738 | 0.7713 | 0.7866 | 0.8031 | 0.8048 |
| solar_flare_m0 | 0.1140 | 0.1787 | 0.1293 | 0.1654 | 0.1592 | 0.1932 | 0.1567 | 0.1508 | 0.0259 | 0.1582 | 0.1179 | 0.1379 | 0.1741 |
| oil | 0.4823 | 0.5447 | 0.5657 | 0.5627 | 0.5539 | 0.4924 | 0.5544 | 0.5348 | 0.5367 | 0.5497 | 0.5238 | 0.5766 | 0.5714 |
| car_eval_4 | 0.9527 | 0.8325 | 0.9484 | 0.9015 | 0.8940 | 0.8475 | 0.8970 | 0.8798 | 0.8344 | 0.9112 | 0.9470 | 0.9174 | 0.9183 |
| wine_quality | 0.2255 | 0.2913 | 0.2301 | 0.2624 | 0.2828 | 0.2733 | 0.2627 | 0.2474 | 0.1863 | 0.2832 | 0.2533 | 0.2582 | 0.2746 |
| letter_img | 0.8635 | 0.7148 | 0.8498 | 0.7903 | 0.7567 | 0.7641 | 0.7660 | 0.7991 | 0.7971 | 0.8394 | 0.8656 | 0.8471 | 0.8028 |
| yeast_me2 | 0.3055 | 0.3485 | 0.3307 | 0.3517 | 0.3659 | 0.3263 | 0.3417 | 0.3558 | 0.3475 | 0.3239 | 0.3605 | 0.3624 | 0.3869 |
| ozone_level | 0.1595 | 0.3482 | 0.3135 | 0.3249 | 0.3443 | 0.3091 | 0.3230 | 0.3188 | 0.3485 | 0.3039 | 0.2602 | 0.3265 | 0.3400 |
| abalone_19 | -0.0036 | 0.0644 | 0.0744 | 0.0947 | 0.0927 | 0.0826 | 0.0974 | 0.0574 | 0.0502 | 0.0943 | -0.0023 | 0.0972 | 0.1046 |
| **rank** | 9.1304 | 7.6522 | 7.6087 | 5.3913 | 6.1739 | 7.3913 | 6.8261 | 8.8261 | 8.8261 | 7.2174 | 7.1739 | **__4.1304__** | **4.6522** |

Table 8: Classification results on benchmark datasets for multilayer perceptron classifier. F1 score averaged over 10 runs using 5-fold cross-validation is reported. **Best** and **second-best** ranked results are highlighted.

| | Imbalanced | Random | Mixup | SMOTE | B-SMOTE | Safelevel | ADASYN | MWMOTE | DBSMOTE | LVQ | AHC | Simplicial | B-Simplicial |
|---|---|---|---|---|---|---|---|---|---|---|---|---|---|
| ecoli | 0.6060 | 0.6409 | 0.6493 | 0.6563 | 0.6301 | 0.6381 | 0.6346 | 0.6552 | 0.6812 | 0.6389 | 0.6679 | 0.6743 | 0.6550 |
| optical_digits | 0.9687 | 0.9654 | 0.9657 | 0.9666 | 0.9657 | 0.9705 | 0.9668 | 0.9659 | 0.9632 | 0.9656 | 0.9709 | 0.9654 | 0.9671 |
| satimage | 0.6728 | 0.6622 | 0.6594 | 0.6708 | 0.6532 | 0.6543 | 0.6616 | 0.6525 | 0.6615 | 0.6562 | 0.6717 | 0.6651 | 0.6607 |
| pen_digits | 0.9885 | 0.9922 | 0.9879 | 0.9912 | 0.9902 | 0.9913 | 0.9910 | 0.9900 | 0.9895 | 0.9888 | 0.9902 | 0.9899 | 0.9911 |
| abalone | 0.0162 | 0.4027 | 0.4385 | 0.4044 | 0.4096 | 0.4070 | 0.3961 | 0.4147 | 0.3988 | 0.3778 | 0.3813 | 0.4175 | 0.4195 |
| sick_euthyroid | 0.7822 | 0.7273 | 0.7629 | 0.7320 | 0.7483 | 0.6546 | 0.7193 | 0.7063 | 0.7609 | 0.7748 | 0.7908 | 0.7739 | 0.7657 |
| spectrometer | 0.8326 | 0.8135 | 0.8029 | 0.8028 | 0.8348 | 0.8269 | 0.8244 | 0.8132 | 0.8159 | 0.8367 | 0.8217 | 0.8276 | 0.8259 |
| car_eval_34 | 0.9632 | 0.9666 | 0.9697 | 0.9651 | 0.9664 | 0.9525 | 0.9696 | 0.9706 | 0.9660 | 0.9599 | 0.9759 | 0.9735 | 0.9727 |
| us_crime | 0.4642 | 0.4857 | 0.4880 | 0.5052 | 0.4879 | 0.5119 | 0.4890 | 0.4860 | 0.4784 | 0.4941 | 0.4839 | 0.4993 | 0.5066 |
| yeast_ml8 | 0.0642 | 0.1025 | 0.1050 | 0.1193 | 0.1200 | 0.1356 | 0.1134 | 0.1104 | 0.0855 | 0.0907 | 0.0751 | 0.1291 | 0.1049 |
| scene | 0.2654 | 0.2536 | 0.2695 | 0.2645 | 0.2801 | 0.2835 | 0.2442 | 0.2582 | 0.2451 | 0.2529 | 0.2567 | 0.2635 | 0.2773 |
| libras_move | 0.8461 | 0.8055 | 0.8466 | 0.8497 | 0.8480 | 0.7955 | 0.8377 | 0.8486 | 0.8428 | 0.8644 | 0.8368 | 0.8373 | 0.8645 |
| thyroid_sick | 0.7244 | 0.7167 | 0.7142 | 0.7161 | 0.7258 | 0.5893 | 0.7159 | 0.6887 | 0.7173 | 0.7248 | 0.7191 | 0.7220 | 0.7235 |
| coil_2000 | 0.1232 | 0.1607 | 0.1564 | 0.1557 | 0.1625 | 0.1678 | 0.1647 | 0.1597 | 0.1298 | 0.1411 | 0.1354 | 0.1509 | 0.1621 |
| arrhythmia | 0.1893 | 0.2562 | 0.2744 | 0.2815 | 0.2288 | 0.3580 | 0.2764 | 0.1996 | 0.2606 | 0.2491 | 0.2565 | 0.3072 | 0.2751 |
| solar_flare_m0 | 0.0770 | 0.1879 | 0.1649 | 0.1599 | 0.1528 | 0.2074 | 0.1694 | 0.1498 | 0.0951 | 0.1627 | 0.0888 | 0.1521 | 0.1579 |
| oil | 0.5539 | 0.5396 | 0.5394 | 0.5402 | 0.5461 | 0.4782 | 0.5463 | 0.5324 | 0.5158 | 0.5634 | 0.5393 | 0.5422 | 0.5900 |
| car_eval_4 | 0.9569 | 0.9563 | 0.9588 | 0.9446 | 0.9589 | 0.9373 | 0.9609 | 0.9565 | 0.9534 | 0.9214 | 0.9752 | 0.9599 | 0.9539 |
| wine_quality | 0.2375 | 0.3065 | 0.2241 | 0.2841 | 0.3175 | 0.2706 | 0.2883 | 0.2493 | 0.2782 | 0.2437 | 0.2866 | 0.2833 | 0.2994 |
| letter_img | 0.9695 | 0.9652 | 0.9537 | 0.9667 | 0.9616 | 0.9544 | 0.9622 | 0.9622 | 0.9566 | 0.9597 | 0.9691 | 0.9636 | 0.9611 |
| yeast_me2 | 0.2450 | 0.3790 | 0.3008 | 0.3549 | 0.3951 | 0.3218 | 0.3635 | 0.3528 | 0.4064 | 0.3448 | 0.4042 | 0.3628 | 0.4105 |
| ozone_level | 0.2527 | 0.2581 | 0.2605 | 0.2679 | 0.2846 | 0.2865 | 0.2900 | 0.2786 | 0.2513 | 0.2722 | 0.2658 | 0.2982 | 0.2924 |
| abalone_19 | 0.0000 | 0.0572 | 0.0642 | 0.0621 | 0.0532 | 0.0483 | 0.0626 | 0.0579 | 0.0400 | 0.0492 | 0.0000 | 0.0677 | 0.0609 |
| **rank** | 8.5870 | 7.4348 | 7.6957 | 6.3043 | 6.0435 | 7.1739 | 6.0000 | 8.1304 | 9.2174 | 8.0435 | 6.7609 | **5.1739** | **4.4348** |

Table 9: Classification results on benchmark datasets for multilayer perceptron classifier. Matthew's correlation coefficient averaged over 10 runs using 5-fold cross-validation is reported. **Best** and **second-best** ranked results are highlighted.

| | Imbalanced | Random | Mixup | SMOTE | B-SMOTE | Safelevel | ADASYN | MWMOTE | DBSMOTE | LVQ | AHC | Simplicial | B-Simplicial |
|---|---|---|---|---|---|---|---|---|---|---|---|---|---|
| ecoli | 0.5737 | 0.6143 | 0.6208 | 0.6284 | 0.5995 | 0.6126 | 0.6051 | 0.6265 | 0.6531 | 0.6139 | 0.6372 | 0.6439 | 0.6236 |
| optical_digits | 0.9655 | 0.9618 | 0.9621 | 0.9631 | 0.9621 | 0.9674 | 0.9633 | 0.9623 | 0.9593 | 0.9621 | 0.9680 | 0.9619 | 0.9636 |
| satimage | 0.6435 | 0.6289 | 0.6248 | 0.6368 | 0.6196 | 0.6258 | 0.6273 | 0.6158 | 0.6297 | 0.6193 | 0.6364 | 0.6291 | 0.6247 |
| pen_digits | 0.9873 | 0.9914 | 0.9866 | 0.9903 | 0.9891 | 0.9903 | 0.9900 | 0.9889 | 0.9884 | 0.9876 | 0.9891 | 0.9888 | 0.9902 |
| abalone | 0.0194 | 0.3821 | 0.3904 | 0.3774 | 0.3810 | 0.3834 | 0.3729 | 0.3852 | 0.3710 | 0.3434 | 0.3170 | 0.3771 | 0.3823 |
| sick_euthyroid | 0.7627 | 0.7050 | 0.7405 | 0.7098 | 0.7259 | 0.6356 | 0.6977 | 0.6815 | 0.7375 | 0.7522 | 0.7698 | 0.7511 | 0.7424 |
| spectrometer | 0.8250 | 0.8013 | 0.7896 | 0.7882 | 0.8253 | 0.8166 | 0.8129 | 0.7999 | 0.8050 | 0.8252 | 0.8124 | 0.8163 | 0.8149 |
| car_eval_34 | 0.9605 | 0.9645 | 0.9678 | 0.9630 | 0.9644 | 0.9500 | 0.9678 | 0.9687 | 0.9640 | 0.9573 | 0.9743 | 0.9717 | 0.9710 |
| us_crime | 0.4285 | 0.4471 | 0.4476 | 0.4666 | 0.4477 | 0.4774 | 0.4494 | 0.4463 | 0.4379 | 0.4541 | 0.4505 | 0.4600 | 0.4696 |
| yeast_ml8 | 0.0299 | 0.0405 | 0.0335 | 0.0547 | 0.0571 | 0.0455 | 0.0425 | 0.0440 | 0.0234 | 0.0333 | 0.0334 | 0.0630 | 0.0421 |
| scene | 0.2348 | 0.1994 | 0.2099 | 0.2100 | 0.2274 | 0.2234 | 0.1889 | 0.2018 | 0.2049 | 0.2043 | 0.2159 | 0.2101 | 0.2278 |
| libras_move | 0.8510 | 0.8047 | 0.8480 | 0.8488 | 0.8482 | 0.7916 | 0.8371 | 0.8485 | 0.8423 | 0.8623 | 0.8415 | 0.8401 | 0.8678 |
| thyroid_sick | 0.7127 | 0.7003 | 0.6969 | 0.6985 | 0.7090 | 0.5801 | 0.6992 | 0.6707 | 0.7002 | 0.7106 | 0.7045 | 0.7052 | 0.7086 |
| coil_2000 | 0.0843 | 0.0970 | 0.0956 | 0.0952 | 0.1036 | 0.1039 | 0.1021 | 0.0994 | 0.0874 | 0.0975 | 0.0847 | 0.0927 | 0.1014 |
| arrhythmia | 0.1513 | 0.2095 | 0.2322 | 0.2418 | 0.1808 | 0.3344 | 0.2342 | 0.1496 | 0.2212 | 0.2049 | 0.2212 | 0.2668 | 0.2433 |
| solar_flare_m0 | 0.0528 | 0.1445 | 0.1185 | 0.1100 | 0.1029 | 0.1705 | 0.1222 | 0.1011 | 0.0570 | 0.1253 | 0.0596 | 0.1089 | 0.1165 |
| oil | 0.5572 | 0.5232 | 0.5237 | 0.5232 | 0.5311 | 0.4627 | 0.5307 | 0.5174 | 0.4974 | 0.5480 | 0.5360 | 0.5289 | 0.5818 |
| car_eval_4 | 0.9570 | 0.9562 | 0.9587 | 0.9446 | 0.9587 | 0.9374 | 0.9610 | 0.9565 | 0.9532 | 0.9213 | 0.9752 | 0.9597 | 0.9537 |
| wine_quality | 0.2667 | 0.2950 | 0.2242 | 0.2716 | 0.2980 | 0.2708 | 0.2781 | 0.2377 | 0.2550 | 0.2134 | 0.2775 | 0.2663 | 0.2749 |
| letter_img | 0.9685 | 0.9642 | 0.9522 | 0.9656 | 0.9603 | 0.9533 | 0.9659 | 0.9609 | 0.9550 | 0.9583 | 0.9680 | 0.9623 | 0.9597 |
| yeast_me2 | 0.2817 | 0.3742 | 0.2996 | 0.3467 | 0.3899 | 0.3303 | 0.3626 | 0.3453 | 0.3933 | 0.3469 | 0.3962 | 0.3546 | 0.3951 |
| ozone_level | 0.2418 | 0.2380 | 0.2403 | 0.2477 | 0.2649 | 0.2815 | 0.2702 | 0.2585 | 0.2309 | 0.2528 | 0.2497 | 0.2796 | 0.2739 |
| abalone_19 | 0.0000 | 0.0633 | 0.0745 | 0.0716 | 0.0506 | 0.0680 | 0.0704 | 0.0548 | 0.0322 | 0.0752 | 0.0000 | 0.0732 | 0.0572 |
| **rank** | 7.7609 | 7.5652 | 8.2174 | 6.6957 | 6.2609 | 6.7391 | 6.3913 | 8.4783 | 9.2609 | 7.4783 | 5.8478 | **5.6087** | **4.6957** |

# C   STATISTICAL SIGNIFICANCE

Table 10: Statistical significance for the F1 score. SMOTE vs Simplicial SMOTE and SMOTE vs Borderline Simplicial SMOTE. Significant p-values of either win or lose are underlined.

| | $k$-Nearest neighbors | | Gradient boosting | | Multilayer perceptron | |
|---|---|---|---|---|---|---|
| | Simplicial | B-Simplicial | Simplicial | B-Simplicial | Simplicial | B-Simplicial |
| ecoli | 1.284e-06 | 7.882e-04 | 8.253e-02 | 3.217e-01 | 9.249e-02 | 9.571e-01 |
| optical_digits | 7.612e-05 | 1.549e-07 | 6.154e-01 | 2.856e-05 | 2.196e-01 | 7.995e-01 |
| satimage | 2.737e-05 | 1.130e-07 | 3.365e-07 | 4.037e-01 | 1.517e-01 | 8.525e-03 |
| pen_digits | 7.211e-03 | 7.221e-04 | 4.517e-02 | 9.068e-10 | 1.101e-03 | 8.233e-01 |
| abalone | 2.011e-07 | 8.020e-06 | 4.000e-05 | 2.027e-05 | 8.776e-06 | 1.253e-05 |
| sick_euthyroid | 1.016e-07 | 2.474e-07 | 3.358e-04 | 2.268e-01 | 5.366e-09 | 7.785e-08 |
| spectrometer | 5.904e-01 | 6.542e-01 | 8.419e-02 | 9.329e-01 | 8.878e-03 | 1.235e-02 |
| car_eval_34 | 6.759e-09 | 3.561e-06 | 3.127e-03 | 4.026e-03 | 5.352e-02 | 2.157e-02 |
| us_crime | 4.533e-05 | 3.367e-09 | 1.159e-02 | 9.816e-03 | 3.722e-01 | 8.853e-01 |
| yeast_ml8 | 5.061e-02 | 4.780e-01 | 8.130e-01 | 3.667e-01 | 3.566e-01 | 2.359e-02 |
| scene | 1.810e-06 | 5.576e-06 | 3.131e-01 | 1.602e-01 | 9.326e-01 | 7.491e-02 |
| libras_move | 1.417e-01 | 9.715e-01 | 5.007e-01 | 7.651e-01 | 1.554e-01 | 3.324e-01 |
| thyroid_sick | 6.381e-09 | 9.002e-09 | 2.113e-01 | 5.534e-02 | 3.320e-01 | 1.720e-01 |
| coil_2000 | 9.961e-01 | 7.334e-02 | 7.557e-10 | 2.310e-05 | 2.113e-01 | 1.126e-01 |
| arrhythmia | 7.670e-01 | 3.914e-01 | 1.000e000 | 9.721e-01 | 3.252e-01 | 5.517e-01 |
| solar_flare_m0 | 6.097e-02 | 4.439e-05 | 5.270e-04 | 4.677e-01 | 2.972e-01 | 4.586e-01 |
| oil | 2.011e-07 | 1.013e-06 | 2.230e-01 | 1.954e-01 | 9.250e-01 | 6.813e-04 |
| car_eval_4 | 1.087e-09 | 1.087e-09 | 5.761e-02 | 1.386e-02 | 3.798e-03 | 7.188e-02 |
| wine_quality | 7.589e-02 | 5.058e-08 | 1.602e-01 | 1.550e-07 | 8.295e-01 | 3.839e-02 |
| letter_img | 8.031e-10 | 8.029e-10 | 7.557e-10 | 2.513e-05 | 5.043e-03 | 1.796e-04 |
| yeast_me2 | 1.913e-04 | 9.637e-08 | 4.517e-02 | 3.679e-06 | 2.997e-01 | 8.774e-06 |
| ozone_level | 4.519e-04 | 3.267e-08 | 8.053e-03 | 8.824e-04 | 5.173e-04 | 1.328e-02 |
| abalone_19 | 3.080e-04 | 1.630e-02 | 1.245e-04 | 1.004e-05 | 3.572e-01 | 7.881e-01 |
| wins | 14 | 17 | 10 | 9 | 5 | 8 |
| losses | 2 | 1 | 2 | 3 | 2 | 3 |
| draws | 7 | 5 | 11 | 11 | 16 | 12 |

Table 11: Statistical significance for the Matthew's correlation coefficient. SMOTE vs Simplicial SMOTE and SMOTE vs Borderline Simplicial SMOTE. Significant p-values of either win or lose are underlined.

| | $k$-Nearest neighbors | | Gradient boosting | | Multilayer perceptron | |
|---|---|---|---|---|---|---|
| | Simplicial | B-Simplicial | Simplicial | B-Simplicial | Simplicial | B-Simplicial |
| ecoli | 3.537e-06 | 4.672e-03 | 1.318e-01 | 2.383e-01 | 1.918e-01 | 5.618e-01 |
| optical_digits | 2.576e-04 | 5.338e-08 | 6.224e-01 | 2.856e-05 | 2.620e-01 | 8.241e-01 |
| satimage | 4.338e-04 | 4.563e-07 | 3.173e-02 | 7.390e-01 | 4.622e-02 | 3.499e-03 |
| pen_digits | 6.563e-03 | 7.055e-04 | 3.662e-02 | 9.068e-10 | 9.440e-04 | 8.451e-01 |
| abalone | 2.638e-03 | 8.824e-04 | 4.661e-01 | 4.037e-01 | 9.643e-01 | 1.630e-01 |
| sick_euthyroid | 2.979e-05 | 3.051e-06 | 9.414e-04 | 3.787e-01 | 1.266e-08 | 1.909e-07 |
| spectrometer | 4.841e-01 | 4.228e-01 | 5.261e-02 | 9.587e-01 | 5.176e-03 | 1.034e-02 |
| car_eval_34 | 1.016e-07 | 4.835e-05 | 4.098e-03 | 6.659e-03 | 6.836e-02 | 2.168e-02 |
| us_crime | 4.749e-03 | 8.308e-07 | 5.061e-02 | 4.314e-02 | 3.299e-01 | 9.643e-01 |
| yeast_ml8 | 2.543e-02 | 5.810e-02 | 6.881e-02 | 1.293e-02 | 4.037e-01 | 5.785e-02 |
| scene | 1.835e-04 | 1.213e-01 | 2.294e-01 | 2.949e-01 | 9.247e-01 | 2.608e-02 |
| libras_move | 2.292e-01 | 9.353e-01 | 5.559e-01 | 5.559e-01 | 3.270e-01 | 2.057e-01 |
| thyroid_sick | 1.633e-07 | 1.495e-05 | 5.399e-01 | 3.085e-01 | 2.986e-01 | 6.452e-02 |
| coil_2000 | 1.962e-01 | 6.958e-01 | 3.679e-06 | 1.307e-02 | 6.328e-01 | 1.478e-01 |
| arrhythmia | 2.444e-01 | 7.995e-01 | 6.784e-01 | 8.888e-01 | 5.254e-01 | 9.115e-01 |
| solar_flare_m0 | 1.652e-01 | 1.800e-04 | 7.334e-02 | 4.901e-01 | 5.921e-01 | 9.269e-01 |
| oil | 1.644e-06 | 3.679e-06 | 4.894e-01 | 3.391e-01 | 4.841e-01 | 1.582e-04 |
| car_eval_4 | 2.229e-09 | 3.784e-09 | 6.748e-02 | 1.608e-02 | 5.324e-03 | 7.852e-02 |
| wine_quality | 1.057e-01 | 6.834e-05 | 2.385e-01 | 2.398e-03 | 4.544e-01 | 5.592e-01 |
| letter_img | 8.031e-10 | 8.532e-10 | 7.557e-10 | 1.570e-03 | 4.894e-03 | 1.796e-04 |
| yeast_me2 | 8.232e-04 | 2.191e-06 | 1.774e-01 | 2.108e-03 | 5.763e-01 | 7.716e-05 |
| ozone_level | 1.479e-02 | 2.737e-05 | 4.314e-01 | 1.573e-01 | 4.214e-04 | 1.214e-02 |
| abalone_19 | 2.527e-06 | 9.730e-01 | 7.246e-01 | 4.544e-01 | 3.616e-01 | 2.113e-01 |
| wins | 15 | 15 | 5 | 7 | 4 | 7 |
| losses | 2 | 1 | 1 | 3 | 2 | 2 |
| draws | 6 | 7 | 17 | 13 | 17 | 14 |

