# OpenReview forum: "Simplicial SMOTE: Oversampling Solution to the Imbalanced Learning Problem"
_ICLR.cc/2024/Conference — ICLR 2024 Conference Withdrawn Submission_

### Official Review · Reviewer_zg3w · 2023-10-29

**Soundness:** 4 excellent
**Presentation:** 4 excellent
**Contribution:** 3 good
**Rating:** 8
**Confidence:** 4

**Summary:**

The paper considers the problem of supervised learning with a class imbalance and proposes a new algorithm for oversampling the minority class based on topological data analysis. The method, Simplicial SMOTE, forms a simplicial complex from data in the minority class, and then generates synthetic samples based on the convex combinations of points sampled from simplices of this simplicial complex. The authors additionally propose variants of other SMOTE algorithms based on simplicial complexes. The utility of the simplicial SMOTE algorithm is validated on synthetic and real datasets, and suggests the value of the approach for a wide variety of empirical applications.

**Strengths:**

The paper is extremely well presented and provides an original application of topological data analysis in a machine learning setting.

The quality of the work is generally high: The empirical results are presented over a wide set of synthetic and empirical datasets with class imbalances which provide a full picture of the proposed algorithm's value.

The proposed algorithm is presented in an extremely clear way and the figures help to highlight why this approach is different than prior methods. Overall, the presentation is excellent and the paper was enjoyable to read.

The work ultimately provides a valuable step in using topological data analysis (TDA) for machine learning: TDA methods are typically computationally intensive (as noted by the authors), and are quick to be dismissed in machine learning applications. However, this paper shows that TDA methods can still add value to empirical performance of learning algorithms and hence provides a foundation for a wide variety of future work.

**Weaknesses:**

Some minor points:

1) Although the paper is quite interesting from the lens of topological data analysis, it is presented as a simplicial extension of SMOTE and hence feels limited in terms of significance for a machine learning audience.

2) The authors could be a bit more clear on why a simplicial complex may be better than a graph for creating synthetic points for oversampling-- it feels like there is some type of local decision-boundary type of argument which would make clear when this method should be valuable.

3) There empirical results could be presented better. The tables are fine and are valuable because they give access to the raw data. However, they could benefit from the addition of confidence intervals. Alternatively, additional visualizations may more clearly summarize the value of the proposed algorithm for each dataset.

**Questions:**

N/A -- it may be helpful if the authors can address point number 2 above.

---

> ### Author Response · Authors · 2023-11-18
>
> Thank you for the high evaluation of our paper, especially noting the use of topological data analysis tools for a general learning problem and the overall quality of experiments and presentation.
>
> Regarding questions:
>
> Q1: The authors could be a bit more clear on why a simplicial complex may be better than a graph for creating synthetic points for oversampling-- it feels like there is some type of local decision-boundary type of argument which would make clear when this method should be valuable.
>
> A1: We assume that a geometric model represented by a simplicial complex is better than one of the graph, i.e. the data distribution is well-approximated by a union of simplices of a geometric neighborhood complex for some set of the neighborhood size parameter k and and the maximal simplex dimension p. We also hypothesize that p could be related to the intrinsic dimension of the data.
> Informally, as we state in the last two paragraphs of introduction, one could not fill the whole data manifold using samples from the edges of the neighborhood graph as edges are 1-dimensional. By increasing the dimension of simplices we fill more space, ideally to match the simplex dimension to the local intrinsic dimension of data. For graphic explanation see https://anonymous.4open.science/r/simplical-smote/motivation.png.
>
> Addressing mentioned weaknesses:
>
> W3: There empirical results could be presented better. The tables are fine and are valuable because they give access to the raw data. However, they could benefit from the addition of confidence intervals. Alternatively, additional visualizations may more clearly summarize the value of the proposed algorithm for each dataset.
>
> A2: Unfortunately, at the time of writing the paper, we did not find a suitable and easy-to-understand way to present confidence intervals and significance testing results between a dozen of methods in a readable and compact way. We would improve the presentation with the critical difference diagrams with pairwise or multiple testing.

---

### Official Review · Reviewer_KN3Z · 2023-11-01

**Soundness:** 3 good
**Presentation:** 3 good
**Contribution:** 2 fair
**Rating:** 3
**Confidence:** 3

**Summary:**

This paper introduces a oversampling technique called Simplicial SMOTE to address
the issue of class imbalance in datasets. Diverging from the conventional SMOTE
technique, Simplicial SMOTE innovates by sampling from simplices of a
geometric neighborhood simplicial complex, as opposed to sampling from the
edges of a geometric neighborhood graph. Through evaluation on the benchmark
datasets, it is concluded that Simplicial SMOTE outperforms the original SMOTE and
its variants.

**Strengths:**

1. Simplicial SMOTE offers an innovative geometric oversampling method to address class imbalance, utilizing topological data analysis tools to extend the capabilities of traditional SMOTE.

2. The method is thoroughly evaluated on a number of benchmark datasets, showcasing its effectiveness and outperformance over existing methods.

3. The paper exhibits clear structure, and high-quality visuals. The writing is clear.

**Weaknesses:**

1. Increased Computational Complexity: The additional steps of computing simplicial complexes and the requirement for parameter tuning (e.g., maximal simplex dimension) could lead to higher computational complexity, potentially limiting the scalability of the Simplicial SMOTE method, especially for high-dimensional or large datasets. This complexity might hinder the real-time or practical applications of the proposed method in scenarios where computational resources or time are constrained.

2. Limited Evaluation on High-Dimensional Datasets: The paper evaluates the proposed Simplicial SMOTE method on 23 benchmark datasets, but it does not provide a thorough evaluation on high-dimensional datasets. The behavior and performance of the method in high-dimensional spaces could be different, and it's crucial to understand how the method scales with dimensionality.


3. Parameter Tuning: The necessity for grid search over the maximal simplex dimension p could be seen as a drawback since it adds an extra layer of complexity to the model tuning process. This could potentially lead to longer setup times before the model can be deployed, especially in a production environment.

4. Overall, the novelty of the proposed method is limited.

**Questions:**

1. How sensitive is the performance of Simplicial SMOTE to the choice of parameters such as the neighborhood size parameter k and and the maximal simplex dimension p ? How were the optimal parameters selected for each dataset in the evaluation?

2. Could the authors provide more insights into the scalability of the proposed method with respect to the number of data points and dimensionality of the datasets? Have the authors tested the method on datasets with larger dimensionality to understand the impact on computational resources and performance?

3. Can the authors provide more details on the computational complexity of Simplicial SMOTE in comparison with the original SMOTE, especially concerning the time and memory requirements? Are there any optimizations suggested to mitigate the computational demand, especially when applying Simplicial SMOTE on large-scale or high-dimensional datasets?

4. Could the authors provide more detailed implementation information to aid reproducibility, such as the specific configurations or hyperparameters used during the evaluation?

5. Have the authors considered comparing Simplicial SMOTE with more recent methods or extensions of SMOTE, other than the ones mentioned in the paper? How does Simplicial SMOTE compare with state-of-the-art methods in handling imbalanced datasets?
How can Simplicial SMOTE be extended to handle multi-class imbalanced datasets? Have the authors considered evaluating the method in multi-class scenarios?

---

> ### Author Response · Authors · 2023-11-18
>
> Thank you for noting the innovativeness of our approach of using topological data analysis tools to address the imbalanced learning problem and the overall clarity of presentation.
>
> Below we added running time showing that our method is only 1.4 times slower than SMOTE, and made publicly available the code of Simplicial SMOTE and the main experiment at https://anonymous.4open.science/r/simplical-smote/. Hope our feedback would allow you to re-evaluate our paper.
>
> Q1: How sensitive is the performance of Simplicial SMOTE to the choice of parameters such as the neighborhood size parameter k and and the maximal simplex dimension p? How were the optimal parameters selected for each dataset in the evaluation?
> Q4: Could the authors provide more detailed implementation information to aid reproducibility, such as the specific configurations or hyperparameters used during the evaluation?
>
> A1: The parameter selection is described in the paper, please refer to the second paragraph of section 4.2. We used the grid search for k with step 2, and for p with step 1 within a range which was dataset-specific, depending on the number of data points and features. You can refer to the parameter selection in the main experiment code  https://anonymous.4open.science/r/simplical-smote/experiment.py.
>
> Q2: Could the authors provide more insights into the scalability of the proposed method with respect to the number of data points and dimensionality of the datasets? Have the authors tested the method on datasets with larger dimensionality to understand the impact on computational resources and performance?
> W2: Limited evaluation on high-dimensional datasets.
>
> A2: We have considered datasets from the popular package imbalanced-learn, with datasets having up to several hundreds of features. The clique expansion stage does not depend on data dimensionality; for the k-nearest neighbors graphs with the same edge density there would not be an impact on the running time of the algorithm.
> One of the assumptions of our method is the manifold hypothesis – despite the data being high-dimensional, described by hundreds or thousands of features, its intrinsic dimensionality is much lower. By the increased dimension of a simplex to sample from, we can better match the intrinsic geometry of the data than SMOTE which models the data as a union of 1-dimensional edges spanned by pairs of existing data points being sufficiently close. For graphic explanation see https://anonymous.4open.science/r/simplical-smote/motivation.png.
>
> Q3: Can the authors provide more details on the computational complexity of Simplicial SMOTE in comparison with the original SMOTE, especially concerning the time and memory requirements? Are there any optimizations suggested to mitigate the computational demand, especially when applying Simplicial SMOTE on large-scale or high-dimensional datasets?
> W1,3: Increased computational complexity due to a single additional hyperparameter (e.g., maximal simplex dimension) and a concern whether this could hinder the real-time or practical applications of the proposed method.
>
> A3: To answer your question whether the method is practical we provide the running time of an experiment on all 23 datasets, with the same pipeline as in the paper including scaler, oversampler and logistic regression as the classifier. We run 5-fold cross-validation repeated 5 times for the neighborhood size parameter k=10. The maximal simplex dimension p was set to 3. Different values of p does not affect computation time as we our current implementation of clique expansion algorithm uses the Bron-Kerbosch algorithm to find all maximal simplices and then subdivide them (which is inexpensive compared to MCE) to p-simplices according to maximal simplex dimension p. Computation was run on 2x Intel(R) Xeon(R) Gold 6248R CPU @ 3.00GHz system, with 48 cores and 96 threads total.
>
> SMOTE, k=10 – 15.03 sec (0.65 sec per dataset on average)
> Simplicial SMOTE, k=10, p=3 – 20.79 sec (0.90 sec per dataset on average)
>
> To conclude, the running time of a typical scenario to fit an oversampler and a linear classifier 25 times is only 1.4 times slower for the Simplicial SMOTE compared to the original SMOTE algorithm.
>
> If a practitioner does not have a time for an extensive grid search of hyperparameters she/he can try to use Simplicial SMOTE with the default value of maximal simplex dimension p, which considers the exact maximal simplices in the neighborhood graph, depending on neighborhood size k, i.e. not limiting their dimension. This in many cases leads to better performance than of SMOTE, yet we obtained the best results by searching for the optimal maximal simplex size. We also observed that simply raising a dimension of a maximal simplex to sample from an edge, as did in SMOTE, to a triangle often resulted in better quality metrics of the downstream tasks. We also plan to release the computationally optimized version of Simplicial SMOTE upon the paper acceptance.

---

> ### Author Response · Authors · 2023-11-20
>
> Q5: Have the authors considered comparing Simplicial SMOTE with more recent methods or extensions of SMOTE, other than the ones mentioned in the paper? How does Simplicial SMOTE compare with state-of-the-art methods in handling imbalanced datasets? How can Simplicial SMOTE be extended to handle multi-class imbalanced datasets? Have the authors considered evaluating the method in multi-class scenarios?
>
> A5: We compared Simplicial SMOTE with classic SMOTE extensions such as Borderline SMOTE, Safe-level SMOTE, and ADASYN, recent methods like Mixup, and the methods taken from the paper [Kovacs19] evaluating 85 SMOTE variants with DBSMOTE, AHC and LVQ-SMOTE shown to get rank 1 for different classifiers and metrics.
> As most of the existing work on SMOTE and its variants we have considered only the binary case. Oversampling methods trivially handle the multi-class scenario, by oversampling all classes except major to match it in size. We would state it explicitly in the updated version of text. Imbalanced-learn, the popular framework implementing SMOTE and several its extensions, contains benchmark data which is binary labeled as well.
>
> [Kovacs19] Kovacs G., An Empirical Comparison and Evaluation of Minority Oversampling Techniques on a Large Number of Imbalanced Datasets

---

> > ### Comment · Reviewer_KN3Z · 2023-11-21
> > **Response to Authors' rebuttal**
> >
> > Thank you very much for your detailed responses to the review comments. I appreciate the time and effort you have put into addressing each point and enhancing the quality of your manuscript. While your replies have greatly clarified many aspects of your manuscript, I still have a few additional questions that I believe would further strengthen the paper if addressed.
> >
> > - The issue of data imbalance is both very practical and critically important. I still believe the authors could experiment with datasets that are closer to real-world scenarios. For example, some standard benchmarks such as Cifar10LT [1,2,3], Cifar100LT [1,2,3], ImageNetLT[3] could be used for this purpose.
> > - As a novel algorithm emerging in the field of data augmentation, Simplicial SMOTE is part of a rapidly evolving landscape of solutions addressing the imbalance problem. To fully assess the contribution and value of Simplicial SMOTE, it is imperative to compare it with state-of-the-art algorithms [4,5,6,7] in this area. Merely comparing it with Mixup and the original SMOTE may not suffice to understand its full potential and positioning in the broader context of current advancements.
> > - If a substantial theoretical analysis of Simplicial SMOTE [8] were conducted, this work would be further solidified.
> >
> > **Considering the time constraints, I would like to see some simple empirical assessments addressing points 1 and 2.**
> >
> >
> >
> > ### Reference
> >
> > 1. Cao K, Wei C, Gaidon A, et al. Learning imbalanced datasets with label-distribution-aware margin loss[J]. Advances in neural information processing systems, 2019, 32.
> > 2. Qin Y, Zheng H, Yao J, et al. Class-Balancing Diffusion Models[C]//Proceedings of the IEEE/CVF Conference on Computer Vision and Pattern Recognition. 2023: 18434-18443.
> > 3. Guo L Z, Li Y F. Class-imbalanced semi-supervised learning with adaptive thresholding[C]//International Conference on Machine Learning. PMLR, 2022: 8082-8094.
> > 4. Berthelot D, Carlini N, Goodfellow I, et al. Mixmatch: A holistic approach to semi-supervised learning[J]. Advances in neural information processing systems, 2019, 32.
> > 5. Sohn K, Berthelot D, Carlini N, et al. Fixmatch: Simplifying semi-supervised learning with consistency and confidence[J]. Advances in neural information processing systems, 2020, 33: 596-608.
> > 6. Mahajan D, Girshick R, Ramanathan V, et al. Exploring the limits of weakly supervised pretraining[C]//Proceedings of the European conference on computer vision (ECCV). 2018: 181-196.
> > 7. Shen R, Bubeck S, Gunasekar S. Data augmentation as feature manipulation[C]//International conference on machine learning. PMLR, 2022: 19773-19808.
> > 8. Elreedy D, Atiya A F, Kamalov F. A theoretical distribution analysis of synthetic minority oversampling technique (SMOTE) for imbalanced learning[J]. Machine Learning, 2023: 1-21.

---

### Official Review · Reviewer_qiAo · 2023-11-01

**Soundness:** 3 good
**Presentation:** 3 good
**Contribution:** 2 fair
**Rating:** 5
**Confidence:** 4

**Summary:**

The paper proposed a generalization of the sampling approach to SMOTE, i.e., sampling from the simplices of the geometric neighborhood simplicial complex. The novelty is the barycentric coordinates concerning a simplex spanned by an arbitrary number of data points being sufficiently close rather than a pair. In the experimental section, the authors evaluate the generalized technique, Simplicial SMOTE, on 23 benchmark datasets.

**Strengths:**

1. The imbalanced classification problem is an interesting and valuable topic in the learning community.

2. The literature part is clear.

3. The structure of the paper is easy to follow.

4. There is extensive experiment analysis on the algorithm performance.

5. The simplicial SMOTE technique can be used to generalize most of the existing types of SMOTE methods.

**Weaknesses:**

1. In the setup section (section 3, p3), it lacks the assumptions and descriptions on the data distribution (x,y), and especially the level of class imbalance. Without data distribution assumptions, it will limit the guidance for practitioners.

2. There is no analysis of the theoretical guarantee of the algorithm's performance.

3. The proposed algorithm is more complicated and slower than the baseline algorithms (see Table 1). However, the time performance of the proposed and baseline algorithms is not shown in the paper. Without time performance, it's hard to judge the tradeoff between time and accuracy in the experimental comparisons.

**Questions:**

1. What's the time performance of the proposed and baseline algorithms?

2. Are there any assumptions and limits on the level of class imbalance?

3. Are there any assumptions on the data distribution to implement the simplicial SMOTE?

---

> ### Author Response · Authors · 2023-11-17
>
> Thank you for noting the problem significance, modularity of our approach, ready to be combined with existing variants of SMOTE, an extensive experimental part and good presentation.
>
> Below we added the running time showing that our method is only 1.4 times slower than SMOTE, and made publicly available the code of Simplicial SMOTE and the main experiment at https://anonymous.4open.science/r/simplical-smote/. Hope our answers would allow you to reconsider the paper evaluation.
>
> Regarding your questions (Q) and stated weaknesses (W):
>
> Q1: What's the time performance of the proposed and baseline algorithms?
> W3: The proposed algorithm is more complicated and slower than the baseline algorithms (see Table 1). However, the time performance of the proposed and baseline algorithms is not shown in the paper. Without time performance, it's hard to judge the tradeoff between time and accuracy in the experimental comparisons.
> A1: Below we provide the running time of an experiment on 23 datasets, with the same pipeline as in the paper including scaler, oversampler and logistic regression as the classifier. We run 5-fold cross-validation repeated 5 times for the neighborhood size parameter k=10. The maximal simplex dimension p was set to 3. Different values of p does not affect computation time as we our current implementation of clique expansion algorithm uses the NetworkX’s Bron-Kerbosch algorithm to find all maximal simplices and then subdivide them (which is inexpensive compared to MCE) to p-simplices according to maximal simplex dimension p. Computation was run on 2x Intel(R) Xeon(R) Gold 6248R CPU @ 3.00GHz system, with 48 cores and 96 threads total.
>
> SMOTE, k=10 – 15.03 sec (0.65 sec per dataset on average)
> Simplicial SMOTE, k=10, p=3 – 20.79 sec (0.90 sec per dataset on average)
>
> To conclude, the running time of a typical scenario to fit an oversampler and a linear classifier 25 times is only 1.4 times slower for the Simplicial SMOTE compared to the original SMOTE algorithm on the benchmark datasets from the imbalanced-learn package.
>
> Q2: Are there any assumptions and limits on the level of class imbalance?
> Q3: Are there any assumptions on the data distribution to implement the simplicial SMOTE?
> W1: In the setup section (section 3, p3), it lacks the assumptions and descriptions on the data distribution (x,y), and especially the level of class imbalance. Without data distribution assumptions, it will limit the guidance for practitioners.
> A2: We assumed that a geometric model represented by a simplicial complex is better than one of the graph, i.e. the data distribution is well-approximated by a union of simplices of a geometric neighborhood complex for some set of the neighborhood size parameter k and and the maximal simplex dimension p. We also hypothesize that p could be related to the intrinsic dimension of the data, and the neighborhood size depends on data’s local geometry.
> We had no specific assumptions on the level of class imbalance (like most of the literature). Common sense argument is that the class imbalance should be at a level where an empirical sample of the minor class is representative for the minor class distribution.
>
> W2: There is no analysis of the theoretical guarantee of the algorithm's performance.
> A3: Indeed, in the paper we have not provided any theoretical analysis of the proposed algorithm. We published the empirical results of classifiers trained on oversampled data, as with most of the papers on SMOTE and its extensions.
> In the process of working on the text, following [Blagus13], we investigated the issue of preserving the mean and variance of data for multivariate Gaussian random variables, but did not consider it new for publication. Also, there was an idea to show that the empirical distribution generated by the proposed algorithm is closer to the ground truth distribution of data in the Wasserstein distance sense, from which we abstained due to text volume limitations imposed by the conference.
> Except [Blagus13], while writing an answer to your question we have discovered several recent papers, studying the theoretical guarantees of SMOTE [Elredy23] and Mixup [Kimura21] methods. We would consider the applicability of the proposed and newer techniques to the theoretical analysis of Simplicial SMOTE.
>
> References
> [Blagus13] Blagus R., Lusa L. SMOTE for High-dimensional Class-imbalanced Data, BMC Bioinformatics (2019)
> [Kimura21] Kimura, M. Why Mixup Improves the Model Performance, International Conference on Artificial Neural Networks (2021)
> [Elredy23] Elredy D., Atiya A., Kamalov F. A Theoretical Distribution Analysis of Synthetic Minority Oversampling Technique (SMOTE) for Imbalanced Learning, Machine Learning (2023)

---

> > ### Comment · Reviewer_qiAo · 2023-11-23
> >
> > Thanks for your detailed response. Most of my concerns are addressed. I will consider them in the next discussion stage.
> >
> > For the running time issue, here is my comment
> >
> > I appreciate your effort to add the code and average time. The average time per dataset is 0.65 seconds and 0.90 seconds for SMOTE and simplicial SMOTE. This means that these datasets are relatively small. The complexity of the simplicial technique could be significant when the data size is large. Therefore, comparisons on a large data set is valuable.

---

### Official Review · Reviewer_mVsN · 2023-11-07

**Soundness:** 3 good
**Presentation:** 2 fair
**Contribution:** 2 fair
**Rating:** 3
**Confidence:** 4

**Summary:**

The paper presents a new extension of SMOTE algorithm for binary imbalanced data that instead of generating new instances as linear interpolations, samples from barycentric coordinated from the constructed simplicial complex. The approach is experimentally tested on 23 real datasets and 4 artificial ones. The results show that variants of the proposed method obtains the lowest average ranks on MCC and F1 measures.

**Strengths:**

- The use of topological analysis (TA) to expand SMOTE is interesting and original. It's probably one of the first works using TA for imbalance problems
- The proposed extension can be also applied to other methods than original SMOTE e.g.  ADASYN, Safe-level SMOTE and Border-line SMOTE

**Weaknesses:**

- Lack of motivation. It is hard to say what research questions are being asked and answered. The main motivation is that the original SMOTE uses neighbourhood relation arity = 2 and the proposed method >2. It is not clear why increasing the arity will help to classify imbalanced data better. Therefore, it's difficult to assess how the paper advances our understanding of the imbalanced learning problem (other than proposing a slightly better performing method).
- The method is currently not suitable for handling multi-class data which is more frequent in practice. The method is tested only on binary classification problems.
- The paper claims to extend several other SMOTE variants like ADASYN, Safe-level SMOTE and Border-line SMOTE but only the latter is actually tested in the experiments.
- Related works.
     - The authors divide the imbalanced learning methods into the rather non-standard three types of methods 1) cost-sensitive 2) under-sampling 3) over-sampling. This categorisation does not include the methods that combine oversampling with undersampling or specialized ensemble methods. - see the book of Fernández at al. "Learning from Imbalanced Data Sets"
  - In the description of the SMOTE method, which is at the focus of this paper, the authors only say that "the new synthetic points as the random convex combinations of pairs consisting of a point and its nearest neighbor", leaving out the rather important information that only neighbours from the same class are considered. This also makes the comparison of Neighbourhood Size/Relation arity in Table 1 a bit misleading, since e.g. Mixup and SMOTE are compared, but the former uses neighbours from the whole dataset and the latter only from the selected class.
  -  In general, the original Mixup (used in the paper) is not a technique addressing class imbalance. Therefore, the Table 1 providing motivation for the approach by comparing it to others, actually compares the proposed approach with SMOTE, non-imbalance learning technique Mixup and Random Oversampling (which do not really use any neighbourhood graph)
  - The authors divide the data level approaches into local (like ROS) and global (like SMOTE). It's quite difficult to understand what local and global mean in this context. Typically, in the imbalanced learning literature, ROS would be a global method (as it focuses on global class imbalance) and SMOTE would be a more local approach (taking into account the local characteristics of a sample). The term "geometric sampling methods" is also newly introduced by the authors and is used, among other things, to refer to random oversampling that do not take geometric relationships into account.
- Clarity. The concept of "sampling from the edges of the neighborhood graph" introduced in the introduction is not very clear to me, and it is not cited. In general, I find the two last paragraphs of the introduction quite difficult to read and the paper would benefit from some more intuitive description.
- Experiments
     - It's hard to say what was the purpose of the experiment on artificial data, since it is not used to observe specific properties of the approach or demonstrate that the method addresses some issue in an isolated environment. The only conclusion relies on an assumption that "circle inside a circle" is more geometrically complex than "two moons" or "swiss rolls"  which I find questionable.
  - The statistical comparisons are performed only with respect to the original SMOTE from 2002 and not to any of the more modern extensions
  - Lack of evaluation with some state-of-the-art implementation of GBT algorithm like CatBoost or XGBoost
  - F1 score metric has been criticized in imbalanced learning literature. It'd be better to also report G-mean value and other specialized metrics.
  - Making the implementation of the method available would increase the reproducibility of this research and its potential impact.


Typo: "first approximating is using a set of prototype points obtained by LVQ"

**Questions:**

- How the datasets were selected? The imbalanced-learn library contains 27 datasets but only 23 were selected for the experiments.
- What was the running time of the proposed method and how it compares with other methods under study?
- The authors use Mixup in their experiment which interpolates not only the feature values x but also target vectors y. Such interpolated target vectors are easy to use in NN but not necessarily in GBT or kNN used in the experiments. How it was applied? If no target vector interpolation was used, how the target value was established?

---

> ### Author Response · Authors · 2023-11-18
>
> Thank you for a very thorough review, we find your comments very valuable, letting us improve the paper. We also thank you for noting the originality and modularity of our approach, ready to be combined with existing variants of SMOTE.
>
> Below we added the running time showing that our method is only 1.4 times slower than SMOTE, our motivation behind the algorithm, and made the code of Simplicial SMOTE and the main experiment publicly available at https://anonymous.4open.science/r/simplical-smote/. Hope it would allow you to re-evaluate our paper.
>
> Regarding questions:
>
> Q1: How the datasets were selected? The imbalanced-learn library contains 27 datasets but only 23 were selected for the experiments.
> A1: For the reason of the speed of an extensive evaluation of 12 oversampling methods using the hyperparameter grid search, we have selected datasets from the imbalanced-learn package which are not exceeding 20000 data points.
>
> Q2: What was the running time of the proposed method and how it compares with other methods under study?
> A2: Below we provide the running time of an experiment on 23 datasets, with the same pipeline as in the paper including scaler, oversampler and logistic regression as the classifier. We run 5-fold cross-validation repeated 5 times for the neighborhood size parameter k=10. The maximal simplex dimension p was set to 3. Different values of p does not affect computation time as we our current implementation of clique expansion algorithm uses the NetworkX’s Bron-Kerbosch algorithm to find all maximal simplices and then subdivide them (which is inexpensive compared to MCE) to the p-simplices according to maximal simplex dimension p. Computation was run on 2x Intel(R) Xeon(R) Gold 6248R CPU @ 3.00GHz system, with 48 cores and 96 threads total.
>
> SMOTE, k=10 – 15.03 sec (0.65 sec per dataset on average)
> Simplicial SMOTE, k=10, p=3 – 20.79 sec (0.90 sec per dataset on average)
>
> To conclude, the running time of a typical scenario to fit an oversampler and a linear classifier 25 times is only 1.4 times slower for the Simplicial SMOTE compared to the original SMOTE algorithm on the benchmark datasets from the imbalanced-learn package.
>
> Q3: The authors use Mixup in their experiment which interpolates not only the feature values x but also target vectors y. Such interpolated target vectors are easy to use in NN but not necessarily in GBT or kNN used in the experiments. How it was applied? If no target vector interpolation was used, how the target value was established?
> A3: In the introduction we stated that we consider the sampling approach used in Mixup (which is generally presented as the data augmentation algorithm) for oversampling of the minor class, without label interpolation. Therefore, we upsampled only the data's minor class, with the label for a synthetic point assigned to be the same as the minor class. Thank you for highlighting that it is not completely clear, it is helpful to us to further improve the text.

---

> ### Author Response · Authors · 2023-11-20
>
> W1: Lack of motivation. The last two paragraphs of introduction are unclear.
> A4: We were motivated by the assumption that a geometric model represented by a simplicial complex is better than one of the graph, i.e. the data distribution is well-approximated by a union of simplices of a geometric neighborhood complex for some set of the neighborhood size parameter k and and the maximal simplex dimension p. We also hypothesize that p could be related to the intrinsic dimension of the data, and the neighborhood size depends on data’s local geometry.
> As we state in the last two paragraphs of introduction, one could not fill the whole data manifold using samples from the edges of the neighborhood graph as edges are 1-dimensional. By increasing the dimension of simplices we fill more space, ideally to match the simplex dimension to the local intrinsic dimension of data. For the graphic explantion see https://anonymous.4open.science/r/simplical-smote/motivation.png
>
> W2: The method is currently not suitable for handling multi-class data, tested only on binary classification problems.
> A5: Our method can be used to handle the multi-class scenario, by oversampling all classes except the major to match it in size. Yet, as most of the existing works on SMOTE and its variants, we considered only the binary case. We should state it explicitly in the updated version of text.
>
> W3: The paper claims to extend several other SMOTE variants like ADASYN, Safe-level SMOTE and Border-line SMOTE but only the latter is actually tested in the experiments.
> A6: We have reported the evaluation results of the simplicial extension of Borderline SMOTE due to the fact that the original Borderline SMOTE performed better than Safe-level SMOTE and ADASYN (see Tables 3, 4, 6 and 7). Moveover, the original Borderline SMOTE was either the best or the second-best among non-simplicial oversampling algorithms, and its simplicial version was shown to always perform better than the original one.
>
> W4: Related works comments related to Mixup and ROS, and the geometric sampling methods taxonomy
> A7: Thank you for highlighting that whether we upsample only the minor class via SMOTE or Mixup (which is generally presented as the data augmentation algorithm) is not completely clear. For the former we implied that it is known for SMOTE practitioners and the latter needs a more explicit explanation. We used Mixup for the oversampling, as it is widely known, improves the solution of the imbalanced learning problem, and fits well to the family of geometric sampling algorithms.
> In topological data analysis there is a concept of filtration of a metric space, where given all pairwise distances we can view them as edges of a graph and add them one by one in increasing order. ROS and Mixup correspond to the extremes of the filtration, with ROS being the zero step and Mixup the last one. SMOTE corresponds to a graph somewhere in between of the filtration, and Simplicial SMOTE to the clique complex of that graph. Next, one decides from simplices of which dimension to sample: 0-simplices for ROS, 1-simplices for Mixup and SMOTE, 2-simplices and greater for Simplicial SMOTE.
>
> W5: Artificial data experiment purpose is unclear
> A9: The purpose of the artificial data experiment was to show the importance of local sampling to respect data’s geometry and topology, which makes SMOTE-based algorithms better than Mixup. In all four examples if the sampling is not local, the synthetic points of the minor class shown in red would either mix with the points of the major class or would result in completely different topology of the decision boundary like in the circle inside the circle example.
>
> W6: The statistical comparisons are performed only with respect to the original SMOTE from 2002 and not to any of the more modern extensions
> A10: Indeed, we showed that both our methods perform significantly better than the original SMOTE, by providing two single-page tables of p-values for each of the used metrics. Unfortunately, we did not find a suitable and easy-to-understand way to present significance testing results between a dozen of methods in a readable and compact way. We will be kindly obliged if you could recommend something in this direction.
>
> W7: Lack of evaluation with some state-of-the-art implementation of GBT algorithm like CatBoost or XGBoost
> A11: During our experiments we have tried Catboost as a classifier, which while resulting in slightly higher values of evaluation metrics (in most cases, the difference was insignificant), but was much slower, so we decided to resort to the implementation of gradient boosting algorithm from the scikit-learn library.

---

> ### Author Response · Authors · 2023-11-20
>
> W8: F1 score metric has been criticized in imbalanced learning literature. It'd be better to also report G-mean value and other specialized metrics.
> A12: Thank you for the suggestion. We used F1 score and MCC as the complementary pair of metrics. F1 score emphasizes the correct classification of the minor class, while MCC considers all elements of the contingency table like G-mean does. We cite the references describing the advantages of MCC in our paper [Chicco20,23].
>
> W9: Making the implementation of the method available would increase the reproducibility of this research and its potential impact.
> A13: We had to obtain permission to make the code publicly-available, and it took more time than expected. For now, everything is OK, and we have our method implemented in Python, made available at an anonymized repository https://anonymous.4open.science/r/simplical-smote/. We plan to release the computationally optimized version of Simplicial SMOTE at publicly-available github repository upon the paper acceptance.
>
> References
> [Chicco20] Chicco D., Jurman G. The Advantages of the Matthews Correlation Coefficient (MCC) over F1 Score and Accuracy in Binary Classification Evaluation. BMC Genomics (2020)
> [Chicco23] Chicco D., Jurman G. The Matthews Correlation Coefficient (MCC) Should Replace the ROC AUC as the Standard Metric for Assessing Binary Classification. BioData Mining (2023)

---

> > ### Comment · Reviewer_mVsN · 2023-11-22
> >
> > Thank you very much for your detailed responses, which helped to clarify some points and allowed me to better understand your approach. Please find below my responses and additional comments.
> >
> > A1. Given that your evaluation of a SMOTE extension on all datasets takes 15/20 seconds, I think you could add some larger (and also higher dimensional) datasets to your study without making it computationally prohibitive.
> >
> > A3. Thank you for the clarification. So is your Mixup just SMOTE with k=n (size of the dataset)?
> >
> > A4. Thank you, I think it would be useful to add this piece of motivation to the paper. This motivation could be further validated in an experiment on artificial data, e.g. you mentioned that you plan to "show that the empirical distribution generated by the proposed algorithm is closer to the ground truth", which is a good idea that would make your paper stronger.
> >
> > A5. This is true, and can be done with virtually any method, but it also somehow goes against your original motivation that the data distribution will be better approximated by simplices. Your method will generate such simplices independently for each class, ignoring the possible overlap between classes. Without an evaluation on multi-class data, it is rather hard to tell to what extent it would influence the results negatively.
> >
> > A6. I think it is quite important to provide the evaluation of new methods presented in a paper, even if you think they will not get the highest score. First, if the extensions are not worth evaluating, then perhaps we should not bother proposing them. Second, even if one of the SMOTE variants performs worse than the other, there is no guarantee that this will hold for its simplicial extensions.
> >
> > A10. Such statistical methods are discussed in the paper https://www.jmlr.org/papers/volume7/demsar06a/demsar06a.pdf Especially, take a look at Friedman test with Nemenyi post-hoc analysis, the results of which can be summarised in a plot (Fig 2).
> >
> > A13.
> > I must admit that I haven't had enough time to thoroughly analyse and run your code, so please correct me if I'm wrong. It seems that you used GridSearchCV to do the hyperparameter tuning and then directly used the mean of the obtained scores for the best parameter values as your final result. This is somewhat problematic because you're selecting the best working hyperparameters and not evaluating them later on a separate test set. Therefore, your results may be overly optimistic.
> >
> > The correct approach would be to do a grid search and later use e.g. a test set. The fact that Grid Search internally uses CV to evaluate the performance of each hyperparameter value does not make the results cross-validated (to achieve this, Grid Search should be run in each iteration of CV).
> >
> > I also think that a README file briefly explaining the contents of each of the repo files would be helpful for potential users.
> >
> > Given the time constraints, could you respond to A13 and potentially A10?

---

### Comment · Area_Chair_KBSo · 2023-11-22
**Author-Reviewer Discussion ends soon**

Dear Reviewers and Authors,

The discussion phase ends soon. Please check all the comments, questions, and responses and react appropriately.

Thank you!

Best, AC for Paper #6425

---

### Author Response · Authors · 2023-11-23

We thank all reviewers for the thorough reviews and comments. We are super excited by the feedback and find it very valuable to improve the paper. Due to the short time frame we are unable to update the paper right now and will improve it on matters highlighted during the discussion stage in a new version.

The list is not exhaustive in any kind, yet some of the topics are motivation, assumptions on data’s distribution and class imbalance ratio, running time, the multi-class case, high-dimensional and real-world datasets, significance testing, hyperparameter selection, evaluations of the simplicial extensions of SMOTE variants, and reproducibility.

We want to thank again the reviewers for their time and their insightful reviews and comments.